# Analyzing Intersectoral Benefits of District Heating in an Integrated Generation and Transmission Expansion Planning Model

Henrik Schwaeppe [1,*], Luis Böttcher [1], Klemens Schumann [1,2], Lukas Hein [1], Philipp Hälsig [2], Simon Thams [1], Paula Baquero Lozano [1] and Albert Moser [1]

1   Institute of High Voltage Equipment and Grids, Digitalization and Energy Economics (IAEW),
    RWTH Aachen University, Schinkelstraße 6, 52062 Aachen, Germany;
    l.boettcher@iaew.rwth-aachen.de (L.B.); k.schumann@iaew.rwth-aachen.de (K.S.);
    l.hein@iaew.rwth-aachen.de (L.H.); s.thams@iaew.rwth-aachen.de (S.T.);
    paula.baquero@rwth-aachen.de (P.B.L.); a.moser@iaew.rwth-aachen.de (A.M.)
2   Fraunhofer Institute for Applied Information Technology (FIT), Schloss Birlinghoven,
    Konrad-Adenauer-Straße, 53757 Sankt Augustin, Germany; philipp.haelsig@fit.fraunhofer.de
*   Correspondence: henrik.schwaeppe@rwth-aachen.de

**Abstract:** In the field of sector integration, the expansion of district heating (DH) is traditionally discussed with regard to the efficient integration of renewable energy sources (RES) and excess heat. But does DH exclusively benefit from other sectors or does it offer advantages in return? So far, studies have investigated DH only as a closed system or determined intersectoral benefits in a highly aggregated approach. We use and expand an integrated generation and transmission expansion planning model to analyze how the flexibility of DH benefits the energy system and the power transmission grid in particular. First of all, the results confirm former investigations that show DH can be used for efficient RES integration. Total annual system cost can be decreased by expanding DH, due to low investment cost and added flexibility, especially from large-scale heat storage. The high short-term efficiency of heat storage—in combination with electric heating technologies—can be exploited to shift heat demand temporally and, using multiple distributed units, locally to solve electric grid congestion. Although it is unclear whether these results can be replicated in the real world, due to the aggregation and detail of the model, further research in this direction is justified.

**Keywords:** district heating; generation expansion planning; generation and transmission expansion planning; heat storage; integrated planning; linear programming; sector coupling; sector integration; thermal energy storage

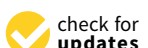



## 1. Introduction

### 1.1. Motivation

Climate change requires radical systemic rethinking and the transformation of the energy system to reduce greenhouse gases. Sector integration, often referred to as sector coupling, promises potentials to reduce greenhouse gas emissions, making intersectoral use of renewable energy and otherwise unused byproducts, such as heat. District heating (DH) is one approach to sector integration which supplies heat for space heating (or cooling), air conditioning, tap water, or manufacturing processes [1]. While today's heating networks are mostly supplied by heating technologies relying on fossil fuels, future systems integrate heat from renewable energy sources (RES), making DH a clean and affordable option under the right circumstances. However, the expansion of DH networks (DHNs) is costly and the question arises whether policymakers should invest more effort in expanding DHNs. This could be reasoned by additional intersectoral advantages, e.g., added flexibility from the heating system to the electric system.

The following paper aims to investigate the advantages of DH in an intersectoral context: offering versatile technology expansion options and therefore the ability to provide flexibility to the electric system. At a high share of renewable electricity, future DHNs could use heat pumps to provide heat [2]. At high electricity demand, cogeneration plants could provide electricity and heat simultaneously. In contrast, *decentral* heating typically uses only one heat source, even though these technologies are also available in small scale. Additionally, DHNs can integrate large-scale heat storages which allow for up to seasonal load shifts and add another degree of flexibility. Heat storages are magnitudes cheaper than other storage options, even by today's standards, and remain to be in the future (compare Table 1) [3]. From an electrical point of view, conventional thermal storage units can only be charged in one direction. However, placed in multiple heating networks and with different schedules, they enable local load shifting of electricity, e.g., for supporting the power transmission grid, but this needs to be further investigated.

**Table 1.** Cost of energy storage in 2050 according to [3].

|  | Large-Scale Hot Water Tanks | Pit Thermal Energy Storage | Lithium-ion NMC Battery (Ulitity-scale) |
|---|---|---|---|
| Energy capacity | 175 $MWh_{th}$ | 4500 $MWh_{th}$ | 8 $MWh_{el}$ |
| Cost (EUR/MWh) | 3000 | 470 | 255,000 |
| Roundtrip Efficiency | 0.98 | 0.70 | 0.92 |

*1.2. State of the Art*

Various approaches have been used and published to evaluate systemic interrelationships in sector integration. Of particular relevance are system studies that examine the flexibility and other advantages of district heating, and system studies which focus on the electric supply and transmission system with respect to heat supply.

The approaches to model and optimize district heating systems are plentiful and have been summarized by Talebi et al. [4]. Most studies investigating the flexibility of district heating in particular neglect the actual heating network and focus on flexibility from an electrical point of view. More specifically, Wang et al. [5], Xu et al. [6], and Yifan et al. [7] focus on current and combined heat and power plants (CHP) to investigate short-term and mid-term flexibility potentials of DH. Kavviadas et al. [8] investigate the combined use of thermal power plants with heat storage and conclude that their combined use improves the overall efficiency and reduces system cost. Lu et al. [9] use an advanced approach to model heating networks with heat and electric supply to investigate the influences of wind generation on the district heating network. However, beyond wind integration, systemic interrelationships are neglected.

Lund et al. [10,11] suggest district heating as one way to integrate RES, balance production, and demand and secure voltage and frequency stability, although also stating that the alternatives need to be investigated. In the dissertation of Gils [12], heat storage is examined as a cheap approach to storing energy from wind peaks. However, the applied REMix model considers energy systems only on a country level. Askeland et al. [13] introduce DH as means of reducing electrical load in Norway, which already has a highly electrified energy system. Bernath et al. [14] investigate the impact of sector integration on the power system and show that the electrification of district heating has a "significant impact on market values", unlike flexible vehicle charging or (decentral) heat pumps. District heating is also considered in the PyPSA-Eur-Sec-30 model by Brown et al. [15]. In their model, power-to-gas, electric vehicles and long-term heat storage contribute largely to smoothing out volatility of RES and, more importantly, to the reduction of overall system cost. Although the PyPSA modeling environment is generally capable of calculating on transmission grid level [16,17], PyPSA-Eur-Sec-30 only models on a country level and therefore neglects parts of the transmission grid. Taking the electrical transmission grid into account, Müller et al. and Metzger et al. [18–20] present energy system studies in which

heating technologies are considered on multiple levels, but technology expansion takes place in a preliminary stage and heating technology is only partially considered for reducing system loads in the operational stage, independent of the transmission grid. On the other hand, Li et al. [21] do not take the expansion of heat storage into account, but operate the transmission grid with respect to the potential flexibility of a district heating network.

To our knowledge, the interplay of district heating and electric transmission networks has not yet been investigated in the light of energy system planning. We identify a research gap in which district heating flexibility is considered from the perspective of reducing system costs and supporting the power grid.

### 1.3. Contribution and Aim of this Study

The aim of this study is to investigate the benefits that district heating can offer to the power transmission grid and, at the same time, what benefits district heating can derive from the electricity system and how it impacts the total system cost from a long-term perspective. For this purpose, we implement district heating in an existing generation and transmission expansion planning (G&TEP) model and analyze the effects in an exemplary scenario in Germany in the year 2045, to understand the impact of DH in the context of a mostly decarbonized future. We describe our implementation of DH into the G&TEP model and describe the data used in Section 2. Using DH penetration and available heat storage capacity as sensitivity, we compare system cost, what technologies are being expanded, and how they are operated in Section 3. Lastly, we discuss our findings and the limits of their validity in Section 4.

### 2. Materials and Methods

The following analysis is conducted using an existing model [22] with an updated framework. The former electric-only G&TEP model is enhanced to integrate thermal demand and additional elements for a more detailed approach to model sector integration. The key elements of the updated framework—heat demand, heat supply technologies, and a simplified representation of district heating—are described in Section 2.1. As input to the model we build up an exemplary German scenario in the year 2045, described in Section 2.2.

### 2.1. Model

The scientific classification of the applied model is a single-stage, purely linear, transmission and generation expansion planning model. The model is implemented in Matlab. In the following, we do not present the model in its full depth, but focus on the functional elements. At first we present the general structure of the existing model and subsequently describe the additionally integrated heating problems.

#### 2.1.1. Generation and Transmission Expansion Planning Model

The presented G&TEP model has originally been conceptualized in [23] and firstly introduced in [22]. The objective of the linear program is to find cost-minimal energy system designs that are compliant with greenhouse gas emission targets and that cover the energy demand of the scenario. Besides decision variables for generation and transmission capacity, the model comprises hourly operational decision variables that allow for the simulation of a full year (8760 hourly time steps). Compared to the model presented in [22], the model has received significant speed improvements through sparse reformulations of storages and the electric transmission grid, which perform significantly better with linear barrier solvers.

Generically, the output/input of an electric technology $i$ at a node $j$ at time step $t$ can be described as $x_{ijt}$, with $x_{ijt} \geq 0$ meaning generation and $x_{ijt} \leq 0$ meaning consumption (active sign convention). Output $x_{ijt}$ is limited by its installed capacity $y_{ij}$. The total expansion potential may further be limited by a constant $Y_{ij}$, such that

$$|x_{ijt}| \leq y_{ij} \leq Y_{ij}. \tag{1}$$

Operation and installation of any technology can be subject to operational cost $C_{ij}^{OPEX}$ and annual investment cost $C_{ij}^{CAPEX}$ that make up the total cost $c_{ij}$; furthermore, operation and expansion of any technology may lead to greenhouse gas emissions $e_{ij}$, with the emission factors described as $E_{ij}^x$ and $E_{ij}^y$, such that

$$c_{ij} = C_{ij}^{CAPEX} \cdot y_{ij} + \sum_t C_{ij}^{OPEX} \cdot x_{ijt} \text{ and} \tag{2}$$

$$e_{ij} = E_{ij}^y \cdot y_{ij} + \sum_t E_{ij}^x \cdot x_{ijt}. \tag{3}$$

Static electric demand at node $j$ and time step $t$ is described by $D_{jt} \leq 0$. Incidence matrix $I_{ij}$ is equal to 1 if technology $i$ is connected to node $j$ and 0 if not. Electric demand and all electric output/input $x_{ijt}$ that is connected to node $j$ describes nodal injection $z_{jt}$ as

$$z_{jt} = D_{jt} + \sum_i I_{ij} \cdot x_{ijt}. \tag{4}$$

In the original model, the transmission grid had been implemented using a PTDF matrix, with a linearized version of virtual injection as suggested in [24]. Virtual injection was removed to increase model performance. For the same reason we use the sparse "Kirchhoff" implementation, as suggested in [25]. Accordingly, nodal injection is used to represent Kirchhoff's current law with $f_{et}$ being the power flow on branch $e$ at time step $t$ and $K_{je}$ being the incidence matrix, describing how branch $e$ is connected to node $j$, as

$$z_{jt} = \sum_e K_{je} \cdot f_{et}. \tag{5}$$

Additionally, a cycle matrix $C_{ec}$, which describes whether branch $e$ is part of a cycle $c$ in a fundamental cycle basis, and the constant reactance $X_e$ of each branch $e$ are used to take Kirchhoff's voltage law into account:

$$\sum_e C_{ec} \cdot X_e \cdot f_{et} = 0. \tag{6}$$

Power flow $f_{et}$ on any branch $e$ should not exceed its defined maximal capacity $F_e$; however, the threshold can be increased through line expansion, or more precisely capacity expansion $f_e^+$, such that

$$|f_{et}| \leq F_e + f_e^+. \tag{7}$$

In addition, $|f_{et}|$ is subject to transmission cost $C_e^f$, indirectly representing losses in the transmission grid, but also tightening the solution space to avoid indifferent optimal solutions. Just as with generation components, grid expansion is subject to expansion costs $C_e^+$ and installation emissions $E_e^f$.

Fuels are considered on an aggregated annual level. There are no time-dependent fuel variables and thus no time-related bottlenecks. The total fuel consumption of *any* fuel by a power producing technology $i$ depends on the technology-dependent efficiency $\eta_{ij}$ and is described by

$$fuel_i^{any} = \sum_t \sum_j -x_{ijt}/\eta_{ij}. \tag{8}$$

Likewise, fuel production, e.g., by electrolysis, would be written as

$$fuel_i^{any} = \sum_t \sum_j -x_{ijt} \cdot \eta_{ij}. \tag{9}$$

Fuel can be imported and exported, described by $fuel_{balance}^{any}$, at the fuel price $C^{any}$:

$$c_{fuel}^{any} = C^{any} \cdot fuel_{balance}^{any}. \tag{10}$$

Imports or exports may be restricted to certain constant limits $Fuel_{imp}^{any} \geq 0$ and $Fuel_{exp}^{any} \leq 0$, as follows:

$$Fuel_{exp}^{any} \leq fuel_{balance}^{any} \leq Fuel_{imp}^{any}. \tag{11}$$

Taken together, *any* fuel is restricted by

$$fuel_{balance}^{any} + \sum_i fuel_i^{any} \geq 0. \tag{12}$$

Taking all parameters into account, the exemplary model is complete and the cost minimization function can be described as

$$min \ \sum_i \sum_j [c_{ij}] \ + \ \sum_e \left[ f_e^+ \cdot C_e^+ + \sum_t |f_{et}| \cdot C_e^f \right] \ + \ \sum_{fuel} c_{fuel}^{any}, \tag{13}$$

where total annual greenhouse gas emissions (*GHG*) should not exceed $GHG_{max}$, therefore

$$\sum_i \sum_j \left[ y_{ij} \cdot E_{ij}^y + \sum_t x_{ijt} \cdot E_{ij}^x \right] \ + \ \sum_e f_e^+ \cdot E_e^f \ \leq \ GHG_{max}. \tag{14}$$

### 2.1.2. Enhancements for Modeling Thermal Demand

The model is further enhanced to integrate problems of thermal demand. Similar to the electrical formulation, thermal demand is aggregated to individual thermal nodes. Heat transmission can be neglected, because it is not economical to transport heat over large distances. Even DHNs usually do not exceed a transmission length of a few dozen kilometers (acknowledging exceptions [26]). Therefore, at the chosen model scale, DH demand can be aggregated to the closest electric node.

The individual, hourly thermal demand at a thermal node $k$ and time step $t$ is described by $H_{kt}$. Each heating node has a constant temperature input and output level that describe the required feed-in temperature and the return temperature after heat exchange. The provided temperature levels must be constant due to the model's linear properties. Temperature levels are, however, used to first of all confirm the right use of a technology and subsequently to determine efficiency parameters. Assuming constant temperature levels for nonlinear thermal processes is a weak premise, but cannot be avoided in a conventional linear program.

Secondly, and similar to electrical generation, we define a thermal variable called $q_{ikt}$, that describes thermal generation or consumption for a technology $i$ at thermal node $k$ at time step $t$. Likewise, thermal technologies may be subject to cost parameters $C_{ik}^q$ and emission parameters $E_{ik}^q$. To be able to distinguish electrical and nodal connections, the formerly introduced investment variable $y_{ij}$ must be enhanced by the thermal node index $k$ and becomes $y_{ijk}$ or $y_{ik}$ if no electric connection is given.

### 2.1.3. Heat Pumps and Electric Boilers

Thermal output of heat pumps depends on the coefficient of performance (COP), described as $COP_{ikt}$. Heat pumps can make use of various heat sources. Air-sourced heat pumps using ambient temperatures are prone to higher efficiency variance than, e.g., ground-sourced heat pumps. Instead of a time-independent COP, we use the methodology suggested in [27] to calculate the time-dependent COP, which is dependent on input and output temperature of source and supply. In case electric boilers are used, the COP is

independent from temperature levels and is immediately set to $0 \leq COP_{ikt} = const. \, \eta \leq 1$. Finally, technology $i$ connected to electrical node $j$ and thermal node $k$ is described as

$$q_{ikt} = -COP_{ikt} \cdot x_{ijt}, \text{ where} \tag{15}$$

$$0 \leq -x_{ijt} \leq y_{ijk}. \tag{16}$$

### 2.1.4. Thermal Energy Storage

As outlined in [28], thermal energy storage (TES) can be used directly (to store higher temperatures and serve lower temperature levels) and indirectly (charging at lower temperatures and using heat pumps to provide to higher temperature levels). We suppose the first variant is used. The most important properties of TES are described by capacity as well as charging, discharging, and standing efficiency. Furthermore, TESs are not suitable for every temperature level. Capacity and standing efficiency are temperature-dependent. Because linear modeling does not allow for nonlinear dependencies, one must decide for a constant temperature level that represents the intended use case or average temperature level, which induces errors inevitably. Modeling parameters should be estimated against worst-case scenarios and results must be considered critically.

The total output balance of (thermal) energy storage is given by its charging and discharging capabilities. As with the electric output, thermal output follows the active sign convention, as follows:

$$q_{ikt} = q_{ikt}^{charge} + q_{ikt}^{discharge}, \tag{17}$$

$$0 \leq -q_{ikt}^{charge} \leq y_{ik} \text{ and} \tag{18}$$

$$0 \leq q_{ikt}^{discharge} \leq y_{ik}. \tag{19}$$

TES, as any energy storage in the model, is implemented using the sparse *stepwise* energy formulation for better solving speed as discussed in [29]. The total energy contained in storage $i$ at node $k$ at time step $t$ is described by $h_{ikt}$. The total charging capacity is described by the energy-to-power ratio $CAP_{ik}$ and the total installed capacity $y_{ik}$ as

$$0 \leq h_{ikt} \leq y_{ik} \cdot CAP_{ik}. \tag{20}$$

The hourly energy balance is described by the energy of the former time step and with respect to the efficiency parameters $0 \leq \eta_{ikt}^{standing}, \eta_{ik}^{charge}, \eta_{ik}^{discharge} \leq 1$ as

$$h_{ikt} = h_{ik(t-1)} \cdot \eta_{ik}^{standing} - q_{ikt}^{charge} \cdot \eta_{ik}^{charge} - q_{ikt}^{discharge} / \eta_{ik}^{discharge} \tag{21}$$

To maintain the energy balance, initial and final energy contained are set to

$$h_{ik0} = CAP_{ik0} \cdot y_{ik} \text{ and} \tag{22}$$

$$h_{ikT} \geq CAP_{ikT} \cdot y_{ik}, \tag{23}$$

with, unless stated otherwise, the initial and final energy-to-power ratios $CAP_{ik0}$ and $CAP_{ikT}$, given as

$$0 \leq CAP_{ik0} = CAP_{ikT} \leq CAP_{ik}. \tag{24}$$

### 2.1.5. Combined Heat and Power

An essential part of DH infrastructures are CHP plants. There is no distinct definition of CHP plants, as technically they may comprise several components. We define CHP plants as technologies which output heat and electricity directly. According to the relevant literature, there are two noncongruent CHP implementations [8,30–32]. Most turbines have a certain power-to-heat ratio, which is described by the so-called backpressure coefficient

$C_{ik}^b$. In the case of backpressure turbines, the power-to-heat ratio is fixed. As also depicted in Figure 1a, heat and electricity output directly depend on each other, so that

$$C_{ik}^b \cdot q_{ikt} = x_{ijt}. \tag{25}$$

In case excess heat can be dissipated, and in the case of steam extraction, heat output is only limited by, and not proportionally equal to, the electrical output. However, the same backpressure coefficient is used to describe this very similar relation, also shown in Figure 1b, as

$$C_{ik}^b \cdot q_{ikt} \leq x_{ijt}. \tag{26}$$

Power plants which extract steam for heat production require an additional parameter $C_{ik}^v$ which describes the loss of electricity when heat production increases by one unit. As a result, the electrical output of a steam extraction plant is limited by $C_{ik}^v$ in such a way that

$$x_{ijt} \leq y_{ijkt} - q_{ijt} \cdot C_{ik}^v. \tag{27}$$

The constraint is also shown in Figure 1b. Non-backpressure CHP plants without steam extraction can use the same model with the parameter $C_{ik}^v = 0$. In such case, Equation (27) becomes an investment constraint, as introduced in Equation (1). In case of $C_{ik}^v \geq 0$, the maximal electric output of power plants is limited by the installed capacity *and* heat output. Lastly, total fuel consumption of BOTH modeling variants is described by

$$fuel_i = \sum_t (x_{ijt} + C_{ik}^v \cdot q_{ikt}) / \eta_{ij}. \tag{28}$$

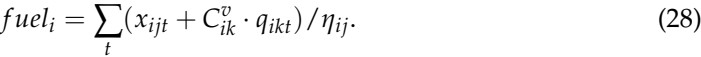

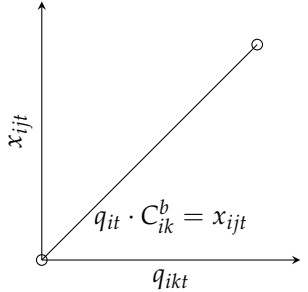
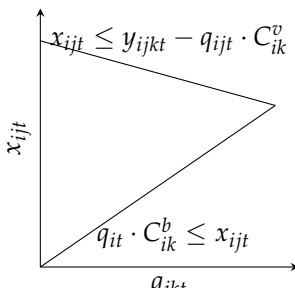

(**a**) Backpressure turbines have a fixed ratio of power and heat output.

(**b**) Extraction condensing turbine ($C_{ik}^v \geq 0$) and further types of CHP plants ($C_{ik}^v = 0$)

**Figure 1.** Different types of heat extraction require different models. Heat output is restricted by $q_{it}$ (heat), $x_{it}$ (electricity) $\geq 0$ and technology-dependent constraints. Power and heat output are independent, but limit each other.

As with any other heating model presented, the nonlinear process of a CHP plant must be modeled in linear fashion. CHP plants, in particular, must also consider minimum power levels and varying startup costs that would require at least some integer variables. However, modeling CHP plants in linear space is common practice. A well-known property of linear optimization problems is to determine corner solutions, e.g., in the case of an extraction condensing turbine there are three corner solutions: no output, only electrical output, and full heat and electrical output (in the apex of the triangle of Figure 1b). Modeling parameters should be selected in such a way that the corner solutions represent outputs close to reality and with respective efficiency parameters [32]. Although two modeling variants have been implemented, in the following we use only the *backpressure turbine* model for CHP plants for simplicity and lack of data on parameters.

### 2.1.6. Modeling District Heating

Two approaches were elaborated for modeling DH, with the second approach proving to be more practical. For *Approach 1*, we used bottom-up energy demand data to determine suitable areas for the expansion of DH. The initial test study was conducted in an area of Northern Germany with a modified IEEE 118-bus test case [33], georeferenced to the area shown in Figure 2. To not overestimate the potential of district heating, the highest available spatial resolution was chosen, as suggested by Jalil-Vega and Hawkes [34]. The used dataset aggregates the energy demand of an estimated 14 million residents into sub-postal code areas, named PLZ8 [35], aggregating to approx. 500 households per code area (detailed data description in Section 2.2.6). It should be mentioned that Dochev et al. [36] suggest that for determining district heating applicability, linear heat density (heat demand per pipe length) is actually more accurate than the used areal heat density, but not even approximate data on linear heat density was available at this scale, e.g., heat demand per street length. According to Möller and Werner [37], an area is suitable for DH expansion if the annual areal heat demand exceeds 8.3 GWh/km². These areas, marked in red in Figure 2, comprise all urban areas and densely populated suburbs, with little to no potential in the east. Respective areas were chosen to be modeled as *DH expansion areas* and aggregated to the closest high-voltage node in the area, with a linear decision variable for DH expansion. The expansion costs per area were varied depending on the areal heat density with the parameters taken from the same source [37], ranging from EUR 40 to EUR 400 per annually distributed megawatt hour. However, to our surprise, and no matter how we parameterized the model, full expansion of all DHNs proved to be the most cost-effective solution of the model. Albeit being an interesting result, this property would not provide much insight, and another approach was chosen.

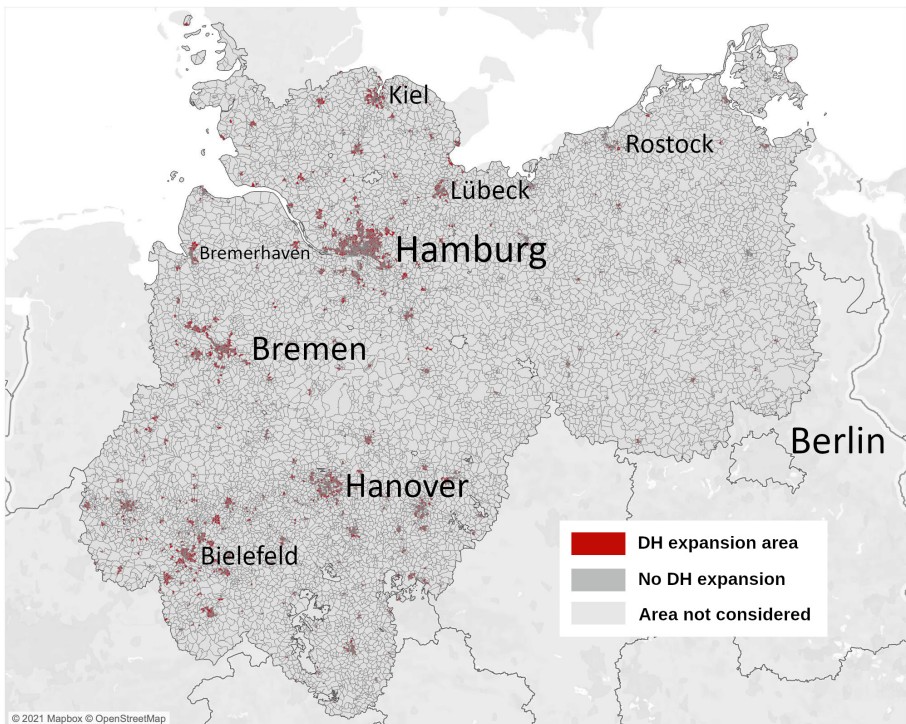

**Figure 2.** *Approach 1* applied in Northern Germany: areas with suitable heat density for DH expansion are marked in red. Some city names have been added for reference. Berlin is outside the area of investigation. A similar dataset, which comprises the whole of Germany, is used in the following.

*Approach 2* defines DH shares upfront. There is no decision variable to decide between DH and non-DH which also reduces linking variables. We use the same bottom-up heating profiles as in *Approach 1*. The DH distribution is derived from power plant data, as public information about DH distribution is sparse (Section 2.2.3). At each electrical node with

DH power plants, we define an additional heat demand which represents DH. In contrast to decentralized heat supplies, DH demand can be be supplied by a variety of technologies, as outlined in Figure 3a. Heating losses from the network must be added to demand *a priori*. Heat losses can be described as a function of supply and return temperature, linear heat density, and ambient temperature [38]. Out of three inputs, only ambient temperature was available. In *Approach 2*, neither linear nor areal heat densities were used to derive potentials, and assumptions about future developments on return and supply temperature were uncertain. It was therefore decided to neglect losses, which was also reasoned by the fact that they were not explicitly modeled with decentral heating options.

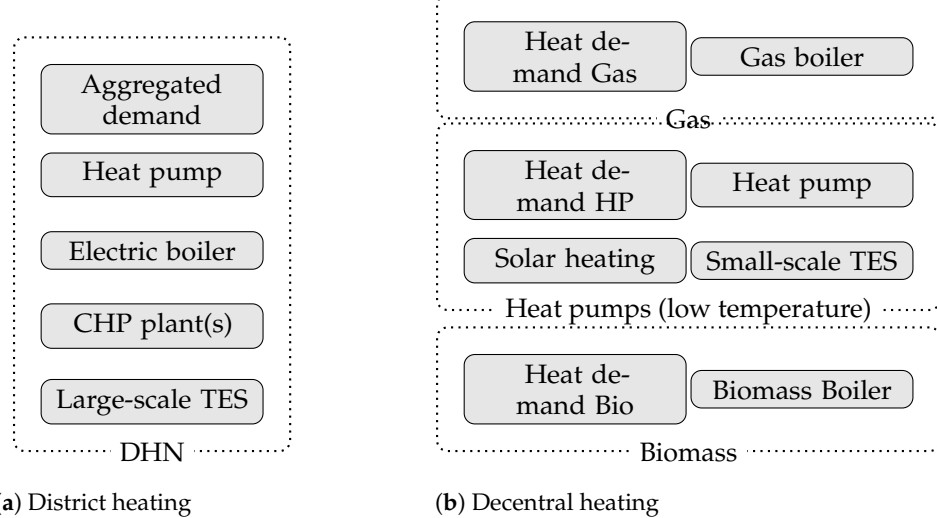

(**a**) District heating      (**b**) Decentral heating

**Figure 3.** Conceptual approach to modeling decentral and district heating on aggregated nodal level.

2.1.7. Modeling Decentral Heating

Decentral heating comprises thermal supply that is not connected to a DHN. It is characterized by commercial and housing units with individual heat supply, e.g., gas boilers or heat pumps. Typically, such units are only provided by one of the available heating technologies. Referring to the case of heat pumps, the technology choice of decentralized heat supply has great systemic impact. It would therefore be obvious to optimize the choice of technology. However, as the share of DH is fixed already, it should also be taken into account that in reality a technology choice is made by individuals and depends on indeterminable contexts. Consequently, optimization is not performed at this point and decentral heating options are determined upfront as a part of the energy system scenario. Decentral heat demand is split into technology-dependent sets of heat demand, as depicted in Figure 3b.

*2.2. Data*

The purpose of this section is to introduce the input data that are used to prepare an exemplary energy system scenario of Germany in the year 2045. The scenario is intended to be representative of a transformed energy system that integrates sector coupling technologies and is consistent with emission reduction targets. Germany was chosen for two reasons: Germany is particularly interesting because a sufficiently large area showcases the interaction between the transmission grid and district heating, and, on the other hand, because the current expansion of district heating networks is not as advanced as in other countries, hence it offers actual expansion potential.

The aim of the following scenario creation is to exemplify applications. We have integrated a variety of data (and methods coupled to them), to represent a future energy system as comprehensively as possible. Required scenario assumptions have been derived from the Agora study *Towards a Climate-Neutral Germany by 2045* [39] and Germany's grid development plan ("Netzentwicklungsplan") for the year 2040 [40]. Data preparation has been designed in line with the requirements explained in [41] and prepared according to the scenario framework outlined in [42]. Data processing has been carried out in the workflow described in [43] and offers the integration of additional datasets, if required later.

### 2.2.1. Focus of Investigation

The following study focuses on a future German energy system, including the German transmission grid and electrical and thermal demand determined bottom-up. Despite having a great impact, neighboring countries are excluded for the sake of demonstration. In terms of time, a post-2045 horizon is envisaged, in which coal and nuclear energy will have been phased out. Based on this, and where applicable, suitable parameters are taken from the study in [39]. Unless otherwise stated, cost parameters are taken from the Danish Energy Agency's technology catalog [3,44] (cost assumptions are provided for 2050). Indirect emission parameters for 2045+ are still hardly obtainable, as discussed in [22]. We use available parameters, e.g., from [45–48], and use plausible assumptions otherwise. Because we also consider indirect emissions (life-cycle assessment), emissions can never be zero. We choose an annual 95% emission reduction target of energy-related sectors from 1990, which we calculated to be 51.8 Mio. t $CO_2$eq. A detailed list of parameters used can be found in Appendix A (Tables A1–A3).

### 2.2.2. Network Data

The base transmission grid model is similar to the one shown in [19], but neglects lines below 220 kV. As means of network reduction, some lines shorter than 25 km are aggregated to one bus. The final grid contains 575 high voltage nodes. The starting grid is a projection of the German transmission grid in the year 2023 and includes AC and DC lines. After aggregation of parallel lines, the number of considered lines totals 802 (compare Figure 4). The capacity on each transmission corridor can be increased by 100%. It allows a direct comparison of whether grid expansion can be avoided between the following sensitivities.

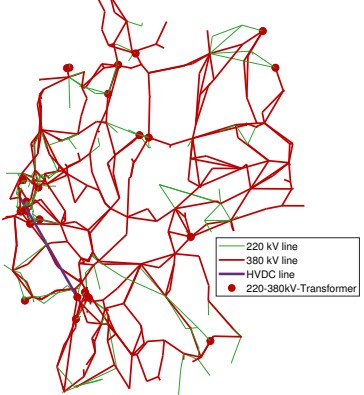

**Figure 4.** Representation of the German transmission grid with 575 busses.

### 2.2.3. Conventional Power Plants

Conventional power plant capacities have been allocated using the Open Power System Data power plant list [49]. With regard to conventional fuel-burning power plants, only those commissioned after 2010 are assumed to be operational in the future scenario. The locations of (decommissioned) coal, gas, nuclear, and oil power plants are used to define locations for potential expansion. Efficiency and cost parameters depend on the type of power plant. We assume that newly-built power plants equate to flexible and

efficient combustion engines. The gas power plant distribution and its expansion potential is displayed in Figure 5a, resulting in a huge potential for the expansion of gas power plants in the densely populated areas in Western and Southern Germany, with only few existing power plants.

The locations of biomass, run-of-river and hydro pump storages are also derived from aforementioned list. No (de)commissioning is expected for pumped hydro storages. Their round-trip efficiency is set to 76%. Locations of run-of-river (RoR) and biomass plants are extracted and scaled to the total capacity of Germany's grid development plan *NEP2021* [40]. The RoR feed-in profiles are constant inputs, totaling the expected energy of scenario B2040. The installed capacity is displayed in Figure 5b. The resulting distribution of RoR and hydro pump storage capacity is very similar to today's distribution. The distribution of biomass power plants is sparser than today, but sufficient for this study.

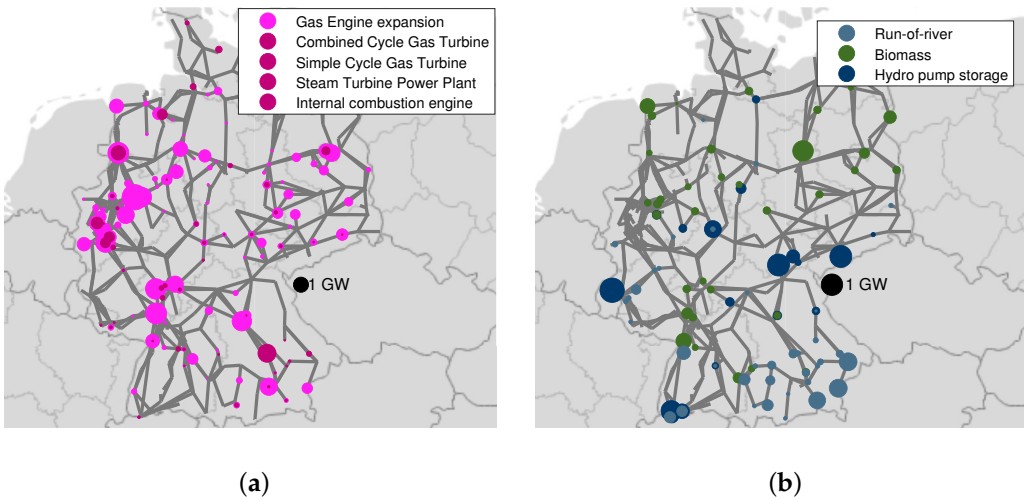

(**a**)                                    (**b**)

**Figure 5.** Installed capacities (and expansion potentials) for conventional technologies. (**a**) Gas power plants and expansion potential. (**b**) Other capacities derived from [49].

### 2.2.4. Renewable Feed-in Profiles and Potentials

As for the possible installation of renewable energy sources, spatial data is used. The potential areas are based on open datasets representing structural and land use information. For the determination of possible sites, a catalog of exclusion sites was determined for each renewable technology. The exclusion sites consider the preservation of natural habitats as well as the minimal distances to inhabited areas. On behalf of technical restrictions, slope and land use information were considered. Figure 6a shows the exemplary results of wind potential areas determined for Germany, with the potentials matched to the nearest node in Figure 6b. Offshore turbines are plotted at their respective connection to the network, but are allocated to the sea for determining their feed-in. The result for photovoltaic expansion is shown in Figure 6c. The feed-in profiles were created using an ERA5 climate dataset [50] in conjunction with the tools provided by *feedinlib* and *windpowerlib* [51,52].

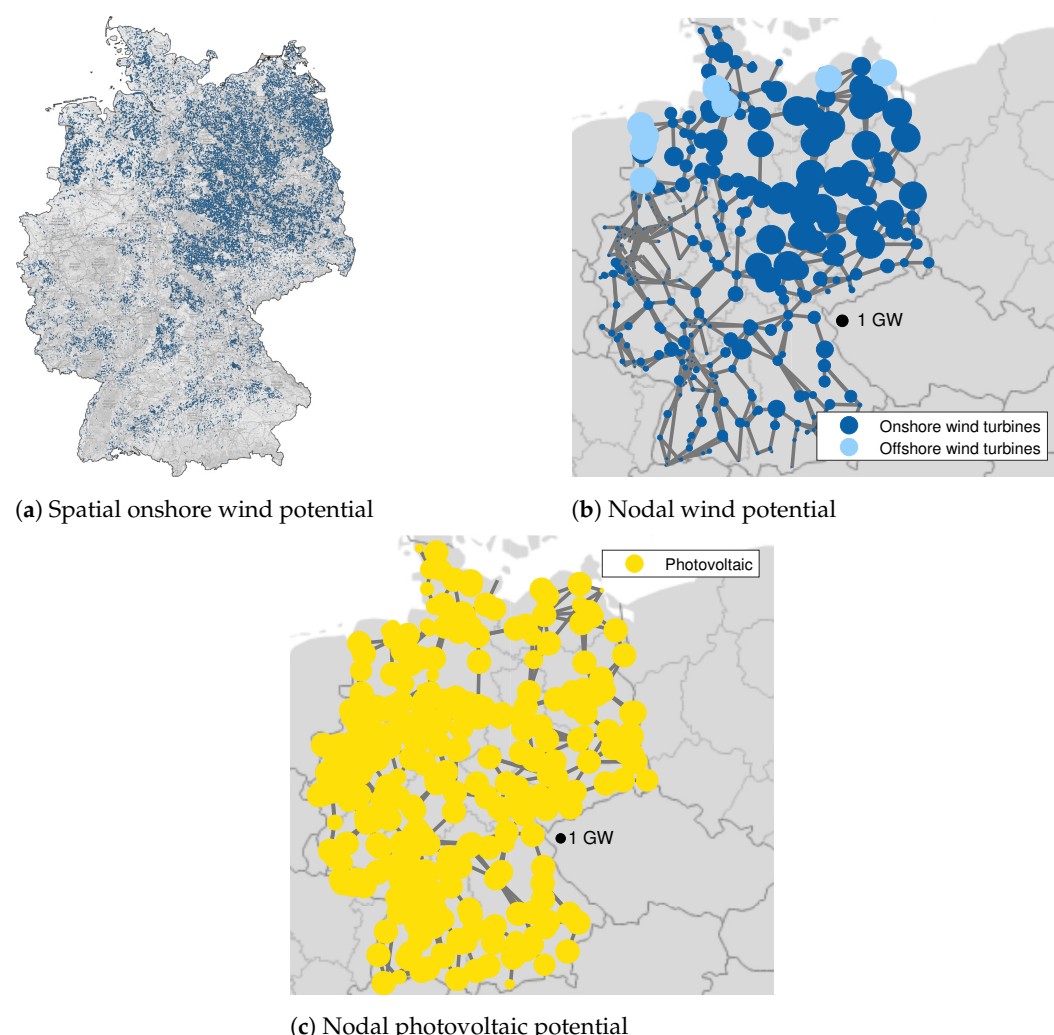

(**a**) Spatial onshore wind potential

(**b**) Nodal wind potential

(**c**) Nodal photovoltaic potential

**Figure 6.** Available potential of variable renewable energy sources on nodal level.

2.2.5. New Technologies

To represent future conditions comprehensively, we integrate some technologies on the rise. As such, we consider combined-cycle power plants with carbon capture storage (CCS), electrolyzers, fuel cells, and battery storage. The power plants with carbon capture are integrated as the more costly alternative to the standard expansion option. Fuel cells do not only output electricity, but can also output heat for district heating. Their locations are the overlapping set of power plant expansion and district heating areas (compare DH section). The required hydrogen can either be imported or produced by the electrolyzers and lastly exported. Based on the assumption that two thirds of hydrogen production will take place in the north of Germany [40], electrolyzers can only be expanded in the north. There are no additional hydrogen demands, except for the model-inherent ones. Although no gas grid is considered, the separation intends to reflect that storage and electricity generation can occur at separate locations. In contrast, batteries can charge and discharge at their installed location. Their locations have been derived from decommissioned coal power plants. The total expansion capacity of electrolyzers and batteries is capped at 3000 MW per node which is above expected expansion. The overall given potential of new technologies can be seen in Figure 7.

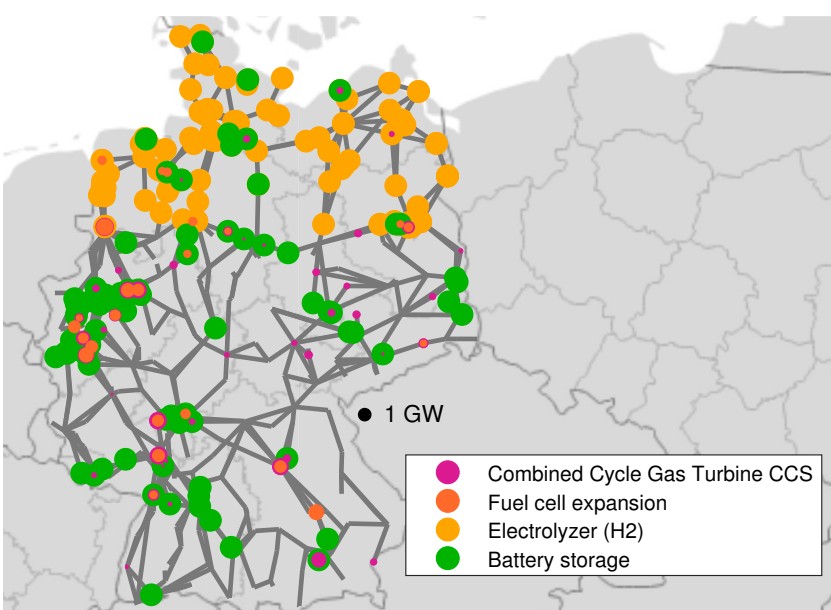

**Figure 7.** New technologies considered in the exemplary scenario.

2.2.6. Bottom-Up Regionalization of Demand Data

Energy demand has been derived bottom-up and includes electrical and thermal demand, as used in [18,53,54]. First, a building database is created storing information about each modeled building in the area of investigation. Second, annual energy demands are determined for each building from which individual time series are generated. The process is depicted in Figure 8.

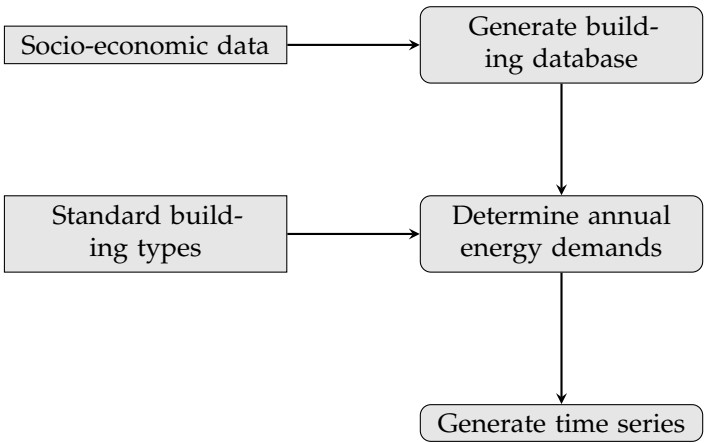

**Figure 8.** Conceptual approach to modeling bottom-up demand data.

The building database is built up using free and licensed socioeconomic data about the number of buildings per construction year and amount of residents for each postal code area [35,55]. The data are blended with the number of employees and businesses per economic department [56]. The final database indicates size, age, and, if applicable, sector and economic department of each building.

Annual energy demand distinguishes between residential and commercial buildings. The energy demand of *residential buildings* depends on size, age, insulation, and the number of residents per building. Size and age are used to assign standard building types with their respective thermal energy demand, given by [57]. The number of residents is used to derive electrical energy demand. The annual energy demand of *commercial buildings* is derived by intersecting socioeconomic properties with statistical data obtained through

the extrapolation of survey data, also provided by [56]. Within, the energy demand for lighting, mechanical work, hot water, space and process heating, process cooling, and communication infrastructure is recorded for 14 different commercial, retail, and service sectors. Using these data, the calculated energy demand depends on the economic department and the estimated number of employees per building.

Time series of thermal demand for *residential buildings* are synthesized with the physical building envelope model described in [58], using the u-values of standard residential building types in [57] and ambient temperature time series from [50]. Their electrical time series are synthesized via the model presented in [59]. We use standard load profiles for *commercial buildings* [60,61]. In principle, demand is modeled on a building-by-building basis. For the purpose of this study, the time series are scaled according to the total energy demand and then aggregated to the locations of respective nodes (of the electrical transmission grid).

### 2.2.7. Energy Demand, and Sensitivities

For the total energy demand, we refer to a study published by Agora [39]. Energy demand is given for several sectors and use cases, but requires adjustments. Energy demand in industry and transportation is assumed to be mostly electrical and is added to the static electric demand of residential and commercial housing (see Figure 9a). Heat demand describes the combination of space heating, warm water, and low-temperature process heat, hence heat demand that can be fulfilled by district heating and is given as useful energy (energy after final conversion, independent of the chosen technologies). Flow and return temperature in DHNs are set to 100 °C/50 °C and 55 °C/25 °C in decentral low-temperature heating. As indicated in Section 2.1.2, temperature levels are used to determine efficiency (of heat pumps) and have no effect on the total energy. Under these conditions, however, decentralized heat pumps are much more efficient.

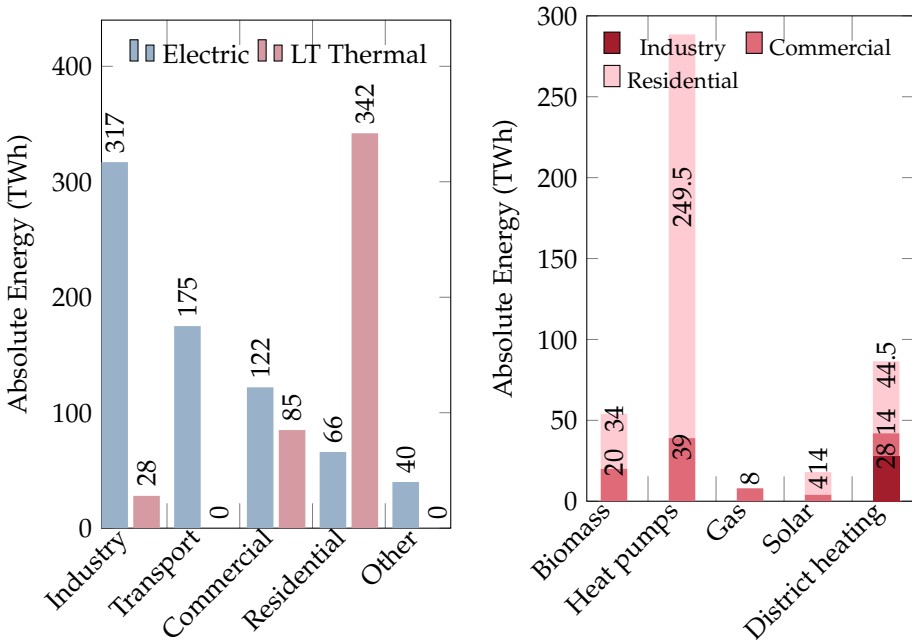

(**a**) Converted energy demand per sector          (**b**) Heat distribution with low DH share

**Figure 9.** *Cont.*

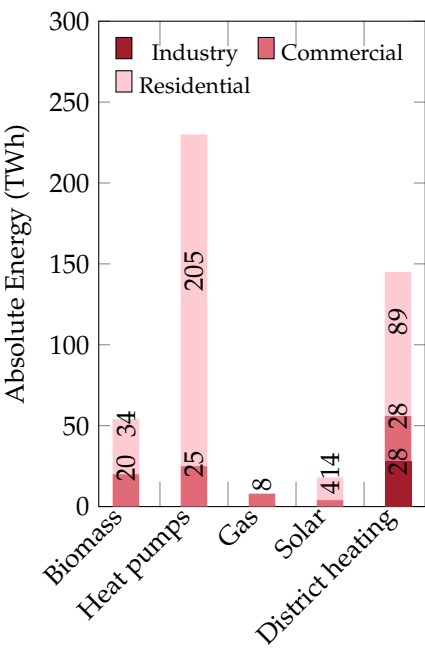

(**c**) Heat distribution with high DH share

**Figure 9.** Static energy demand and heating distribution.

To investigate the systemic benefits of district heating, exemplary sensitivities are carried out. Two parameters are varied: the share of district heating in the system (see Section 2.2.7) and the availability of heat storage in district heating. Depending on the scenario, either 16.6% (Figure 9b) or 27.8% (Figure 9c) of that heat is supplied by DH. The *low DH share* is reminiscent of Germany's current situation and can be seen as the scenario with no additional DH expansion effort. The *high DH share* represents a desired DH expansion with similar values found in [39,62]. Thermal supply capacities in the Open Power System Data power plant list [49] were used to derive in total 118 DH nodes (representations of DHNs). As a base case, heat storage expansion potential per node is limited to a capacity of 3 GWh (or 50 MW input/output). The CHP plant *Küstenkraftwerk KIEL* in Germany can already store up to 1.5 GWh [63], and therefore the chosen limit is within reasonable scale. As a sensitivity, and parallel to the increased DH share, the heat storage expansion potential can also be doubled. Combining the two variations of DH share and heat storage, we obtain four sensitivities. Their respective identifiers are also summarized in Figure 10.

|  | 3 GWh TES | 6 GWh TES |
|---|---|---|
| Low DH share | 2045+ | STOR |
| High DH share | ADDH | BOTH |

**Figure 10.** Names of the sensitivities, varying DH penetration and thermal storage availability.

## 3. Results

### 3.1. Calculation

The sensitivities have been calculated on nodes at the *RWTH Compute Cluster*. Different types of computing nodes have been used, so run time cannot be compared directly. Nevertheless, using 16 cores and not more than 200 GB RAM, it took between 12 and 18 h to retrieve results. Model build-up in Matlab takes less than 3% of the total time. We use Gurobi's barrier implementation with deactivated crossover and a barrier tolerance of $1 \times 10^{-5}$ to solve the models. Although no basic optimal solution is guaranteed, the optimality gap is marginal and is hardly distinguishable from numerical inaccuracies.

### 3.2. Expansion of Technologies Connected to the Power Transmission Grid

An understanding of the general result context is necessary for interpretation. We limit ourselves to the most important results before investigating DH-related results. One important outcome of the model is nodal expansion decisions. Comparing the results of sensitivity 2045+ with both (shown in Figure 11), the system design—or, rather, the installed capacity per node—is found to differ very slightly.

Across all scenarios, about 185 GW of onshore wind is expanded with a focus on the northeastern regions. Similarly, more than 400 GW of photovoltaics is expanded, but with a focus on the southwestern areas and in strong correlation with the expansion of roundabout 57–61 GW (229–245 GWh) of battery storage. Offshore turbines are being expanded to just around 20 GW. The grid expansion measures barely differ. Yet, comparing the extreme scenarios (2045+ and BOTH), the annuity-based network expansion costs are reduced from EUR 267 million to EUR 244 million, indicating a rather small reduction of required transmission capacity. Other technologies diverge more strongly in comparison, which can be related to the performed sensitivities.

In scenario 2045+, onshore wind turbines provide the largest energy share at 484 TWh, followed by photovoltaic with 362 TWh. Offshore turbines provide roughly 83 TWh. In total, 87 TWh of that energy must be curtailed due to lack of demand or lack of transmission capacity. Added together, the three technologies alone provide more than 84% of the total energy demand. In addition to 720 TWh of static demand, there is also energy demand from flexible technologies: 163 TWh from power-to-gas plants, 94 TWh from heat pumps and 22 TWh from electric boilers. The total energy demand sums up to around 1000 TWh of electric energy demand. Flexible technologies such as batteries, heat pumps, boilers, and power-to-gas plants make use of surplus renewable energy. While the charge of batteries is directly given back to the system, heat pumps and electric boilers make use of provided heat storage to shift their electric load. The produced hydrogen of electrolyzers is used in fuel cells to provide electricity and heat at times of low renewable feed-in. In few cases, gas-fired power plants are used to meet the remaining energy demand. An excerpt of the aggregated electric operation is shown in Figure 12. For the sake of completeness, it should also be noted that 0.05 percent of electrical energy demand cannot be met and is therefore shedded via costly *slack variables*.

Gas-fired power plants are responsible for more than 66% of the overall $CO_2$ emissions, closely followed by grid expansion, which makes up 23% of the annual emission budget. Without discussing the details, there is arguably still potential to further reduce the emission of climate gases. A detailed breakdown of $CO_2$ distributions per sensitivity can be found in Appendix B (Figure A1).

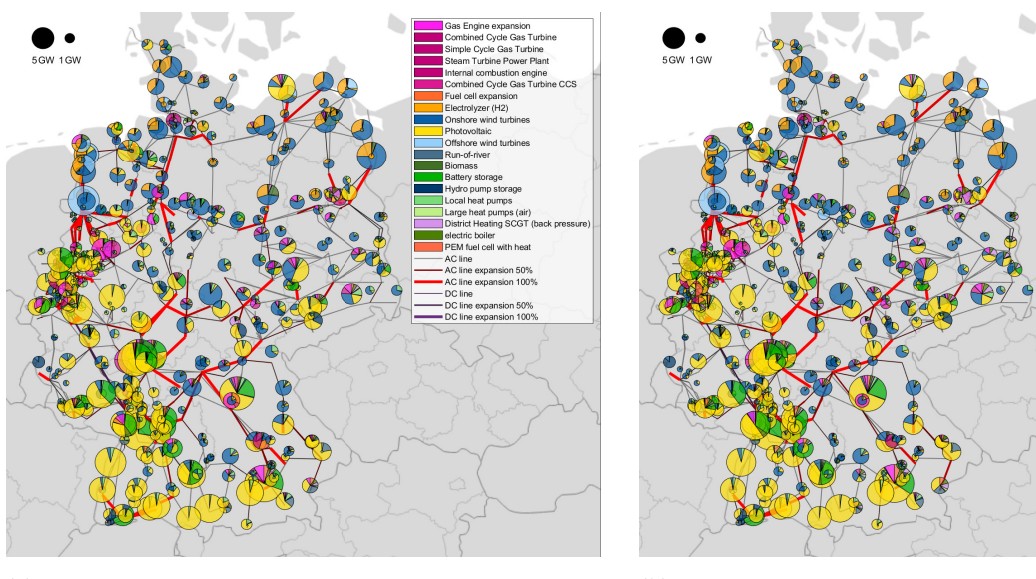

(**a**) Sensitivity: 2045+                                                    (**b**) Sensitivity: BOTH

**Figure 11.** The results of electric technology expansion in two sensitivites only vary slightly and must be compared in greater detail.

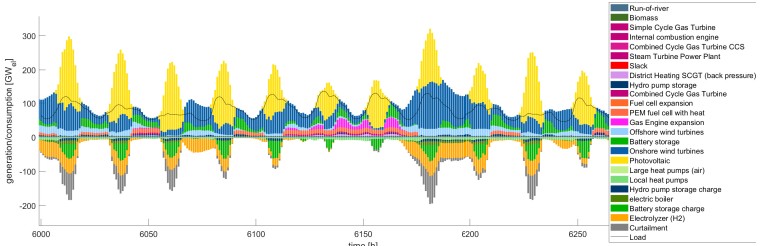

**Figure 12.** Excerpt of the aggregated electric dispatch in scenario 2045+.

### 3.3. Expansion and Operation of District Heating Technologies

The comparison of the sensitivities reveals expectedly larger differences for district heating technologies, as, here, demand has been varied. All available technologies are being expanded, but with regionally varying patterns. As shown in Figure 13, the increased demand of district heating consequently leads to an increase in technology expansion. Interestingly, more expensive fuel cells are preferably expanded in the south, similar to the north–south separation of wind and solar. Power plants in the south run at higher full-load hours to eliminate bottlenecks in the power grid, justifying the use of a more expensive, but also emission-free, technology. Conversely, backpressure turbines in the north have lower full-load hours and meet more peak loads. Thermal storage is being developed to its full potential almost everywhere. The total output capacity is comparatively low in relation to the total district heating capacity, which is, however, very high to provide a diverse portfolio for demand coverage. The expansion of *all four* sensitivites has been plotted in Appendix C (Figure A2).

An excerpt of the aggregated dispatch of district heating demand is shown in Figure 14a, with the thermal dispatch of one particular node in Northern Germany in Figure 14b. The node under study did not expand a heat pump, unlike in Figure 14a. Electric boilers and excess heat from a backpressure turbine are used to charge the heat storage. A fuel cell provides the baseline of the heat demand. Apparently, the operational patterns of the aggregated and nodal perspective do not match, which indicates regional load shifting. We will return to this in the following.

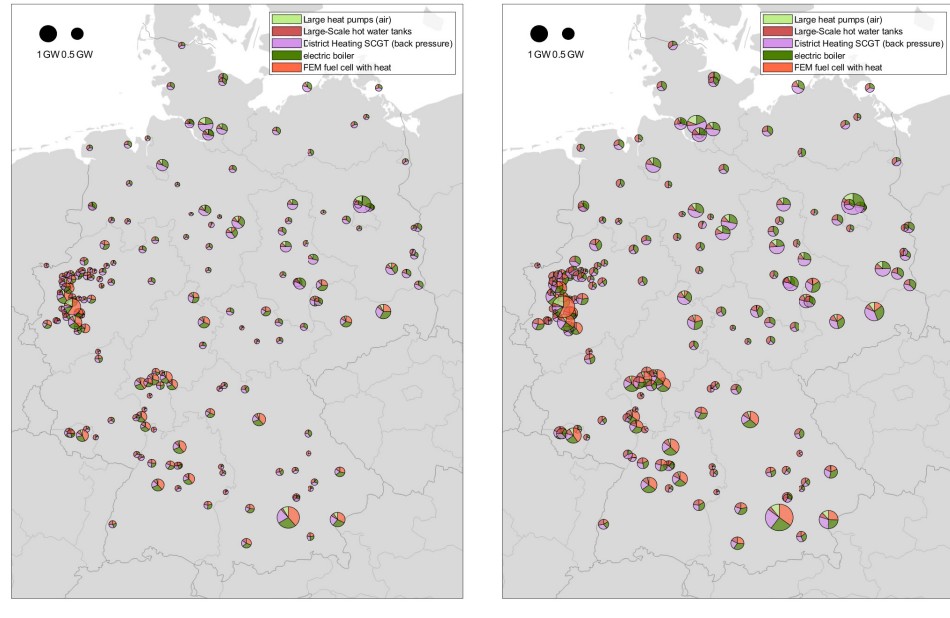

(**a**) Sensitivity: 2045+

(**b**) Sensitivity: BOTH

**Figure 13.** Comparison of district heating technologies with their output capacity: due to higher demand, overall expansion increases.

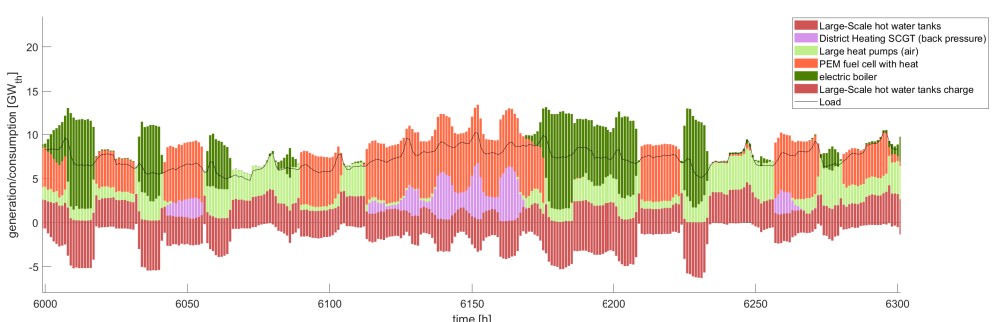

(**a**) Aggregated thermal dispatch of district heating nodes

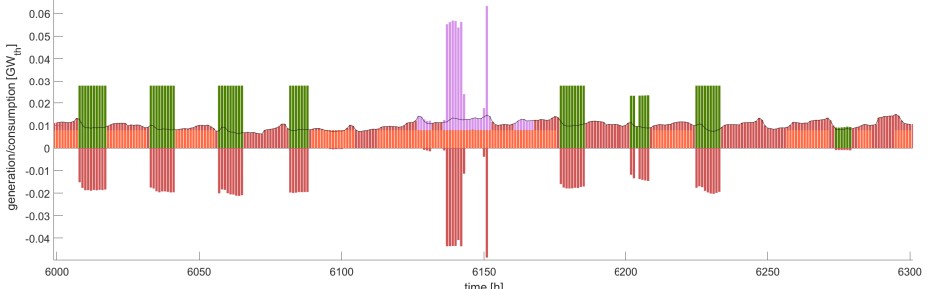

(**b**) Thermal district heating demand of a single node

**Figure 14.** District heating operation in scenario 2045+.

## 3.4. Quantifying Potential Benefits of District Heating

To summarize systemic advantages of district heating, the total annual cost of the presented sensitivities are compared, with the share of district heating in heat supply, as well as the availability of heat storage, being varied. In the baseline scenario *2045+*, it amounts to EUR 38.53 billion. Increasing the potential of heat storages decreases the annual cost down to EUR 38.30 billion. Increasing the share of district heating amounts to 36.89 billion euros. Increasing both, share and storage, totals to EUR 36.67 billion. It indicates that district heating and heat storage independently have a positive impact on

lowering the total cost and do not supersede each other. Some cost-reduction potential can directly be attributed to the fact that instead of cost-intensive decentralized heat pumps, less expensive centralized heating systems are used (in ADDH and BOTH). The values are also summarized in Figure 15.

| 2045+: 38.53 billion EUR | STOR: -0.23 billion EUR |
| ADDH: -1.64 billion EUR | BOTH: -1.86 billion EUR |

(**a**) Total annual system cost (incl. slack cost)

| 52.13 EUR per MWh | -0.79 EUR per MWh |
| -2.64 EUR per MWh | -3.20 EUR per MWh |

(**b**) Average system cost of electricity demand

| 20.99 EUR per MWh | -0.23 EUR per MWh |
| -0.87 EUR per MWh | -1.03 EUR per MWh |

(**c**) Average system cost of heat pump demand

| 3.43 EUR per MWh | +0.08 EUR per MWh |
| +3.40 EUR per MWh | +3.55 EUR per MWh |

(**d**) Average system cost of district heating demand

**Figure 15.** Overview of results from sensitivities. Results of the base scenario are provided as absolute values. The other values are given as deviation from the base scenario.

The cost of providing energy can be indicated by evaluating the dual cost of respective demand constraints, also referred to as the shadow price. The average and energy-weighted cost of increasing energy demand by a fractional unit has been calculated for electricity, heat pump, and district heating demand and is summarized in Figure 15b–d. The values cannot be interpreted as an energy price, since they also comprise technology expansion and other costs; hence, we refer to it as *system cost*. The system costs of electricity demand decrease by more than 6% between 2045+ and BOTH. The same applies to heat pump demand, solely depending on electricity, which decreases by roughly 5%. In contrast, the cost of providing district heat increases by more than 100%, albeit starting from a very low cost level. Since additional flexibility is provided in all sensitivities, the cost reductions for electricity demand are intuitive. The cost increase of district heating needs further explanation.

Figure 16 features three plots with the nodal and energy-weighted average hourly system cost of each energy demand. Nodal costs are plotted in an overlay of partially transparent color. Thus, visibility of color indicates hourly cost variations in the system. The energy-weighted average is plotted in black. To keep the overview, the graphs are truncated to fully plot the energy-weighted average. It can be considered that very high hourly system cost indicates bottleneck work or the elimination of bottlenecks through expansion. The energy-weighted hourly system cost of electricity and heat pumps in Figure 16a,b do not fall below zero with remarkable upward deviations in heating periods. Contrary to this, hourly system cost of district heating falls below zero for several hours in respective time periods. In this case, district heating is the beneficiary of the expansion of district heating power plants whose electrical energy is needed to stabilize the electrical system. Thermal energy is a byproduct; however, the costs induced by the power plants are

attributed to electric demand. Consequently, adding (or shifting) thermal energy demand to district heating results in lowered total system costs in respective hours. At the same time, when demand is added, cost-reduction potentials are reduced, which is why system prices for district heating rise. The explanation is also validated by the heat storage sensitivity, in which additional load can be shifted to considered hours through charging.

Even when neglecting these effects, the systemic costs of DH (excluding heating network costs) are a fraction of the other systems due to upscaled, and therefore cheaper, investment cost, as well as more flexible options. This is confirmed by the fact that, although available, no decentral storage or solar-thermal applications are expanded in any sensitivity.

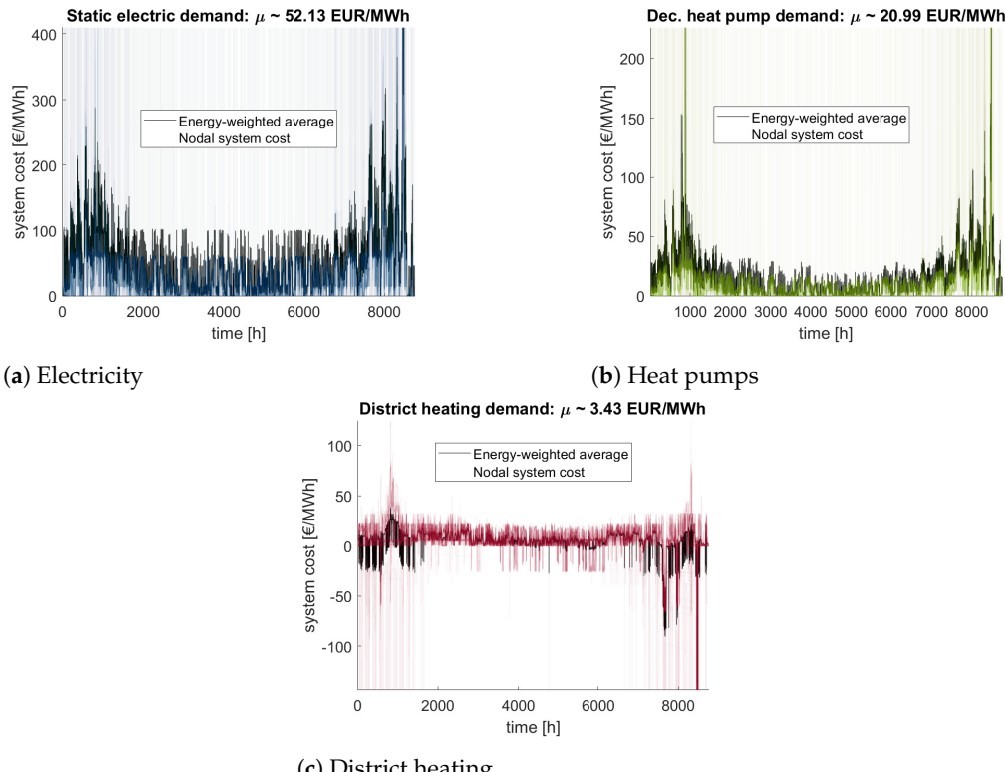

(**a**) Electricity

(**b**) Heat pumps

(**c**) District heating

**Figure 16.** Price curves of nodal demand (in colors) with energy-weighted average in scenario 2045+.

*3.5. Comparison of Different Storage Operations*

Storages have different properties with regard to the energy stored, their size, power, and efficiency. Hydro pump storage cannot be expanded and is limited to today's installed capacity. Depending on the scenario, battery storage is expanded to 57–61 GW or (229–245 GWh). Large-scale hot water tanks connected to district heating are expanded to their full potential (with only few exceptions). Decentral heat storage is not expanded due to its comparably high installation cost (which is caused by its small scale).

A major difference between the electrically connected storage (battery and hydro pump) and heat storage is the energy output in relation to the energy that can be stored. As demonstrated in Figure 17, the stored energy fluctuates strongly over time in electrically connected storage due to the superior energy-to-power ratio and compared to hot water tanks. Battery storage is generally used more frequently than hydro pump storages. Compared to hot water tanks, both operations show some regularity to when they are being used. Heat storages' charging peaks (output $\leq 0$) tend to be larger than their discharging peaks. Trivially, discharging is limited to current heat demand, but charging is not—obviously, surplus electric energy is used in the heating system.

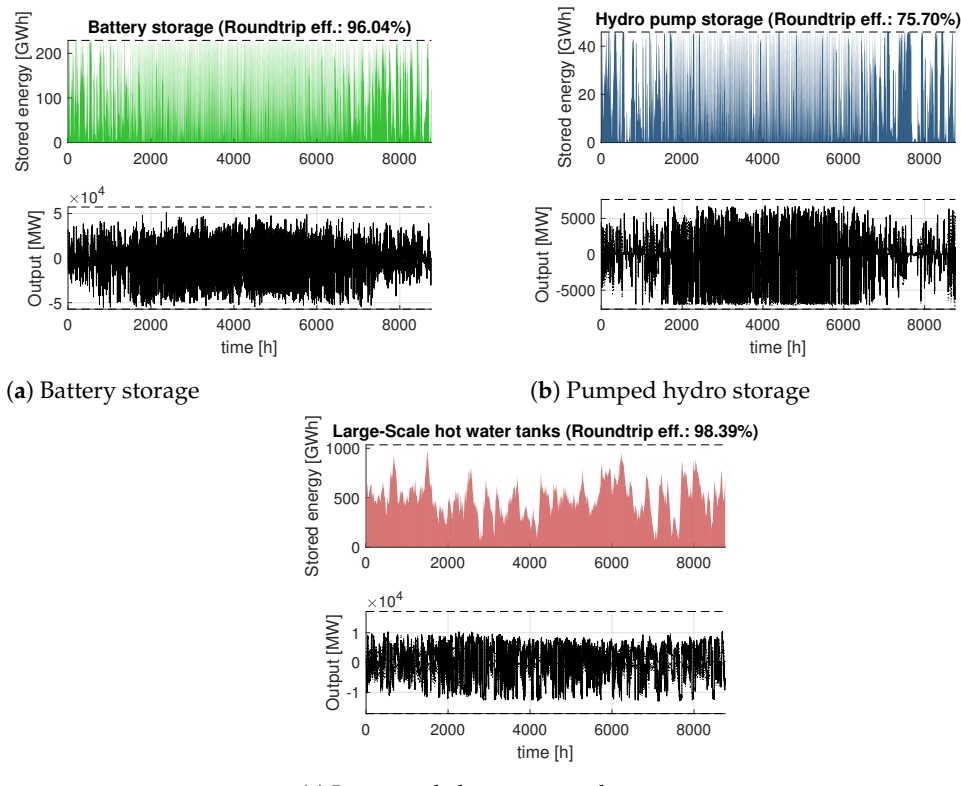

(**c**) Large-scale hot water tanks

**Figure 17.** Stored energy and energy input/output in scenario BOTH. The shown patterns do not vary significantly per scenario, only the magnitude of heat storage changes according to scenario.

A technology-dependent difference between storage operation can also be noticed in individual operation patterns. In Figure 18, we aggregate storage to those that charge and those that discharge energy and compare those values with the total aggregated output to measure the simultaneity of their individual operation. Simultaneity is technology-dependent. Battery storages charge and discharge very simultaneously (Figure 18a) and are predominantly used to compensate for daily fluctuations. Hydro pump storages operate similarly (Figure 18b). Heat storage operation (Figure 18c) is only simultaneous at its extremes. Heat storages charge and discharge permanently at the same time. A large part of this behavior can be explained on the basis of the efficiency and charging performance of the technologies. Apart from very minor standing losses, hot water tanks have minor energy losses in the short term, and their flexibility, as we will explain, is likely used to solve bottlenecks in the transmission grid.

If the operational profiles of heat storages are divided into two clusters using kmeans, the georeferenced profiles divide as shown in Figure 19. Independent of the randomness of the clustering algorithm, the clustered profiles divide the study area between the northeast and southwest. It represents the wind–photovoltaic subdivision that is also representative of the grid bottlenecks (where transmission capacity has been expanded), as shown in Figure 11. Heat storages are thus presumably used to balance grid bottlenecks and fluctuations of renewable energies in the short term and in an energetically sensible way.

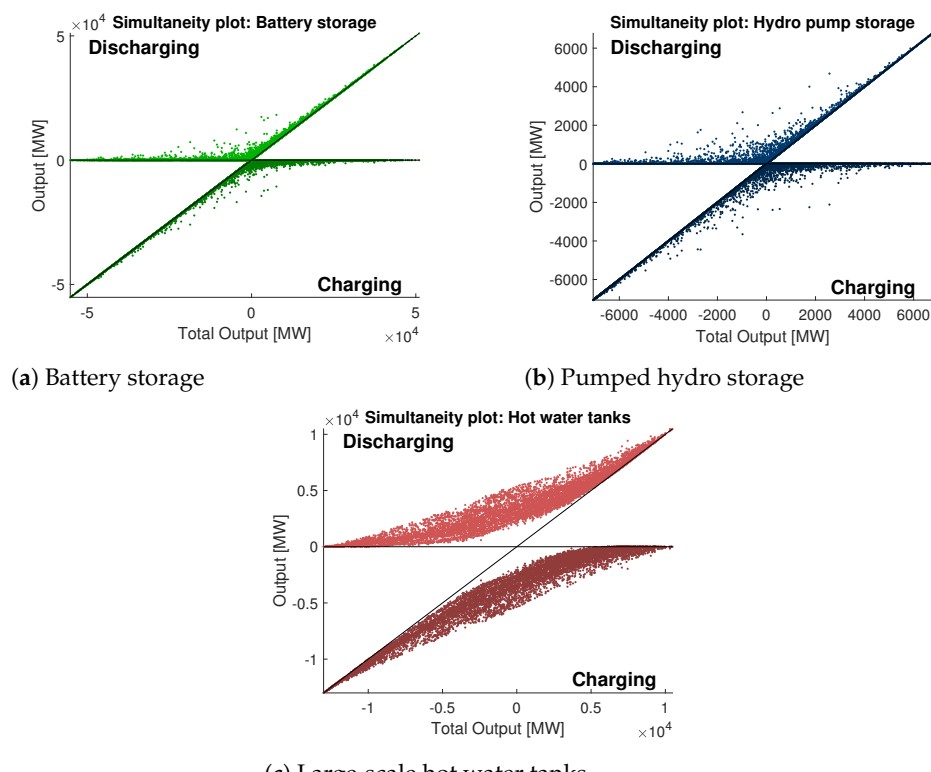

(**a**) Battery storage   (**b**) Pumped hydro storage

(**c**) Large-scale hot water tanks

**Figure 18.** Simultaneity of storage operation in scenario BOTH. The shown patterns do not vary significantly per scenario, only the magnitude of heat storage changes according to scenario.

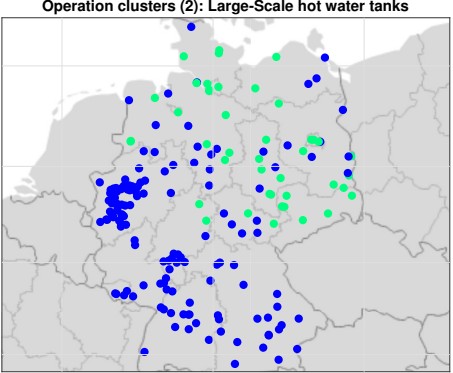

**Figure 19.** Operation clusters: operational time series of hot water tanks clustered in two groups (blue, green) and plotted to their respective location (scenario: BOTH).

## 4. Discussion

As shown by Trutnevyte et al. [64], optimization models are never going to be coherent with the actual future, and this paper does not claim to predict the future of an energy system. Instead, a reasonable future scenario is used to analyze the particular benefits of district heating. In the context of multi-energy system planning models, Hawker and Bell [65] reflect on the usefulness of "optimal pathways" and urge to mitigate "the danger of results being more reflective of design choices than the properties of the real-world systems being modeled". However, the aim of this paper was not to determine final system designs, but to investigate the usefulness of DH on the basis of sensitivities. Inevitably, the problems that arise from huge amounts of data and the corresponding multitude of incorporated model data cannot be avoided, but should remain manageable for the purpose of this paper. All in all, key results of this paper are as follows:

- It does not make sense to model the cost of DHN expansion at the chosen model scale. First modeling attempts indicated that district heating is the preferred heating option and was always expanded to the maximum due to immense cost savings and increased flexibility—at least if expansion was allowed in sufficiently dense areas (compare Section 2.1.6). Apart from sensitivity studies, the informative value of model-endogenous DHN expansion is probably negligible without further knowledge.
- DH offers strong cost reduction and flexibility potential from the electric point of view. This does not yet take into account other effects, such as the integration of excess heat to reduce emissions.
- At least within the given sensitivities, the solution, apart from DH, changed only slightly. Although the total system cost is reduced, the rest of the energy system does not need to be specially adapted for this purpose. In other words: district heating can be integrated smoothly.
- Under the right circumstances, large-scale heat storage in district heating enables better RES integration which reduces cost and adds flexibility to the coupled electric system.
- Enhancing district heating networks (and thus implicitly their demand) simultaneously increases the potential of large-scale heat storage.
- At least within the parameter space of the sensitivity study, DHN expansion does not significantly reduce the need for high-voltage grid expansion and should rather be understood as an element to provide short-term flexibility.

In interpreting these statements, one must bear in mind that the cost of DHN expansion has not been considered and must be determined exogenously. Although the indicated annual cost reduction potential is well over EUR 1 billion *without* the cost of expansion, it should be sufficient to argue for the expansion of district heating networks; actual costs and savings could be in balance in reality. Another point is that the flexibility potential of district heating networks and storage with highly nonlinear properties has been assessed in a purely linear model and, although countermeasures have been applied, could still be overestimated. Regardless, the results are consistent with, and support the findings of, existing studies (sufficient RES integration [10,11], heat storages to store the energy of wind peaks [12], smoothing out volatile RES production [15]), now incorporating opportunities with respect to the electric transmission grid. An obvious shortcoming of this study is that neighboring countries were not included. Taking them into account would presumably reduce the cost of countering bottlenecks and result in lower total system cost. Another shortcoming is the neglect of heating losses in DHNs. The added heat demand requires more heat production and would presumably reduce the "storage potential", as storage could be used to respond to the heating losses. The overall effect of DH and heat storage must be re-evaluated under more realistic conditions to fully understand the consequences.

There is also some evidence that the potential could be underestimated. The available storage size per district heating grid has been chosen liberally and offers potential for improvement. Negative system cost for heat supply indicates that there still is surplus heat due to the operation of CHP plants for electricity supply. In addition, the interplay between various storage options (e.g., battery, hydrogen, and heat storage) has not been analyzed in detail and could be investigated with approaches of modeling to generate alternatives, as suggested in [17].

The required operations of district heating and heat storage, in particular to the benefit of the power transmission grid, are very complex and can only be accomplished in a holistic planning and operation model. Today's energy-only markets do not incentivize such a complex operation, and further market signals are required. However, the expected complexity of such flexibility management is not unique to thermal storage and should therefore be treated as a general problem. It remains that large-scale heat storage is cheap and can offer systemic benefits. Even in the light of an overestimation of the technology in this study, large-scale heat storage offers great potential that should be further investigated.

Lastly, this study did not consider future developments of DH, in particular fourth and fifth generation district heating attempts [66]. Whether lowering the operational

temperature and including the purpose of cooling in DH increases or decreases the flexibility potential (to the benefit of the power transmission grid) should be the focus of further studies.

**Author Contributions:** Conceptualization, H.S. and A.M.; methodology, H.S., L.B. and P.B.L.; software, H.S., L.B., L.H., K.S., P.H., S.T. and P.B.L.; validation, H.S., P.B.L. and S.T.; formal analysis, H.S.; investigation, H.S.; data curation, P.H., K.S., L.B., H.S., L.H. and S.T.; writing—original draft preparation, H.S., L.B., L.H., K.S. and S.T.; writing—review and editing, H.S.; visualization, H.S., L.B., P.B.L. and K.S.; supervision, H.S. and A.M.; project administration, H.S. All authors have read and agreed to the published version of the manuscript.

**Funding:** This research is part of the project PlaMES (Integrated Planning of Multi-Energy Systems). PlaMES has received funding from the European Union's Horizon 2020 research and innovation programme under grant agreement No. 863922. The content of this publication reflects only the authors' view. The European Climate, Infrastructure and Environment Executive Agency (CINEA) is not responsible for any use that may be made of the information it contain.

**Acknowledgments:** Development work and prior test simulations that led to the results shown here were performed with computing resources granted by RWTH Aachen University under project ID 4037 and 4359.

**Conflicts of Interest:** The funders had no role in the design of the study; in the collection, analyses, or interpretation of data; in the writing of the manuscript, or in the decision to publish the results.

## Abbreviations

The following abbreviations are used in this manuscript:

| | |
|---|---|
| a | annum (per year) |
| CHP | Combined heat and power |
| CO2 | Carbon dioxide |
| CO2eq | Carbon dioxide equivalent |
| COP | Coefficient of performance |
| d | Day |
| el | Electric |
| DH | District heat(ing) |
| DHN | District heating network |
| G&TEP | Generation and transmission expansion |
| GHG | Greenhouse gas emissions |
| GW | Gigawatt |
| GWh | Gigawatt hour |
| HP | Heat pump |
| MW | Megawatt |
| MWh | Megawatt hour |
| RES | Renewable energy sources |
| RoR | Run-of-river |
| TES | Thermal energy storage |
| TW | Terrawatt |
| TWh | Terrawatthour |
| t | (metric) ton |
| th | thermal |

## Appendix A. List of Used Parameters

We have compiled an exhaustive list of parameters used for the energy system model in Table A1 (heating), Table A2 (electric), and Table A3 (other inputs). In most cases, values not given mean that the parameters are irrelevant (0 EUR/MWh) or they can be derived directly from other values (CO2 emissions based on efficiency). Related sources have been included.

**Table A1.** Parameter table heating.

| Heating Inputs | | | | |
|---|---|---|---|---|
| Name | C | Capacity/Volume | Count | Parameters |
| Heat load DH | | 86.5 TWh/145 TWh | 182 nodes | Section 2.2.7 [39] |
| DH SCGT (backpressure) | | [0 .. 84.5] GW | 182 | $\eta_{el}$: 0.5, $C^b$: 1.0, CAPEX: 38.8k EUR/MW/a, OPEX (excl. fuel): 4 EUR/MWh$_{el}$ [3] |
| DH water tanks | | [0 .. 9] GW$_{th}$/[0 .. 18] GW$_{th}$ | 182 | Section 2.2.7 [39], (dis)charge eff.: 100%, standing losses: 0.3 %/d, energy-to-power-ratio: 60.35, CAPEX: 972 EUR/MW$_{th}$/a, OPEX: 0.1 EUR/MWh$_{th, discharged}$ [44], 0.5 t$_{CO2}$/MW/a |
| PEM fuel cell with heat | | [0 .. 84.5] GW | 182 | $\eta_{el}$: 0.5, $C^b$: 1.25, CAPEX: 120k EUR/MW/a [3], 6 t$_{CO2}$/MW/a |
| Large HP (air) | | [0 .. 10.8] GW$_{el}$ | 182 | CAPEX: 32.4k EUR/MW$_{el}$/a [3], OPEX (excl. el.): 1.7 EUR/MWh/$_{el}$, Supply: 100 °C, Return: 50 °C, 10 t$_{CO2}$/MW$_{el}$/a |
| Electric boiler | | [0 .. Inf] | 182 | $\eta$: 0.99, CAPEX: 3.9k EUR/MW/a, OPEX (excl. fuel): 0.4 EUR/MWh$_{el}$ [3], 2 t$_{CO2}$/MW/a |
| Heat load HP & Solar | | 306.5 TWh/248 TWh | 508 nodes | Section 2.2.7 [39] |
| Local HP | | [0 .. Inf] | 508 | CAPEX: 101 EUR/kW/a$_{el}$/a, 12.5 kg$_{CO2}$/kW/a [67], Supply: 55 °C, Return: 25 °C |
| Decentral water tanks | | [0 .. Inf] | 508 | (dis)charge eff.: 100%, standing losses: 39%/d, energy-to-power-ratio: 0.15, CAPEX: 6.8 EUR/kW/a$_{th}$/a, OPEX: 1.2 EUR/MWh$_{th,discharged}$ [67], 1 t$_{CO2}$/MW/a |
| Solarthermal | | [0 .. 5.3] GW$_{th}$ | 508 | $\eta$: 0.5, CAPEX: 29k EUR/MW/a [3], 1 t$_{CO2}$/MW/a |
| Heat load Gas | | 8 TWh | 508 nodes | Section 2.2.7 [39] |
| Gas boiler | | [0 .. Inf] | 508 | $\eta$: 1.0, CAPEX: 3.6k EUR/MW/a [67], 2 t$_{CO2}$/MW/a |
| Heat load Biomass | | 54 TWh | 508 nodes | Section 2.2.7 [39] |
| Biomass heating | | [0 .. Inf] | 508 | $\eta$: 0.9, CAPEX: 27k EUR/MW/a [67], 2 t$_{CO2}$/MW/a |

**Table A2.** Relevant parameters for technologies not directly related to heating.

| Name | C | Capacity/Volume | Count | Parameters |
|---|---|---|---|---|
| **Other Inputs** | | | | |
| Electric load | | 720 TWh | 575 nodes | Section 2.2.7 [39] |
| CCGT (gas) | | 4538 MW | 16 | $\eta$: 0.56, OPEX (excl. fuel): 4.4 EUR/MWh [3] |
| SCGT (gas) | | 145 MW | 3 | $\eta$: 0.40, OPEX (excl. fuel): 4.4 EUR/MWh [3] |
| STPP (gas) | | 2027 MW | 34 | $\eta$: 0.40, OPEX (excl. fuel): 4.4 EUR/MWh [3] |
| ICE (gas) | | 33 MW | 2 | $\eta$: 0.44, OPEX (excl. fuel): 5.4 EUR/MWh [3] |
| PP expansion (gas) | | [0 .. 42] GW | 125 | $\eta$: 0.48, CAPEX: 42.5k EUR/MW/a, OPEX (excl. fuel): 4.9 EUR/MWh, 4.8 $t_{CO2}$/MW/a [3] |
| CCGT CCS (gas) | | [0 .. 25] GW | 50 | $\eta$: 0.55, CAPEX: 34.8k EUR/MW/a, OPEX (excl. fuel): 75 EUR/MWh, 4 $t_{CO2}$/MW/a |
| Biomass PP (el.) | | 6 GW [40] | 36 | $\eta$: 0.375 |
| Fuel cells (H2) | | [0 .. 16] GW | 24 | $\eta$: 0.5, CAPEX: 120k EUR/MW/a, 6 $t_{CO2}$/MW/a |
| Electrolyzer (H2) | | [0 .. 366] GW | 122 | $\eta$: 0.705, CAPEX: 28k EUR/MW/a, 4 $t_{CO2}$/MW/a |
| Onshore wind | | [55 .. 401] GW | 418 | CAPEX: 38.4k EUR/MW/a [3], 11.2 $t_{CO2}$/MW/a [47] |
| Offshore wind | | [7.8 .. 80] GW | 8 | CAPEX: 59.3k EUR/MW/a [3], 10.9 $t_{CO2}$/MW/a [68] |
| Photovoltaic | | [54 .. 3138] GW | 477 | CAPEX: 15k EUR/MW/a, [3], 5 $t_{CO2}$/MW/a [47] |
| Run-of-river | | 5.6 GW | 42 | 28 TWh [40], Locations [49] |
| Hydro pump storage | | 7.6 GW | 21 | Locations [49], energy-to-power-ratio: 6, $\eta_{roundtrip}$: 0.76 |
| Battery storage | | [0 .. 240] GW | 80 | CAPEX: 35k EUR/MW, OPEX: 1 EUR/MWh$_{discharge}$, $\eta_{roundtrip}$: 0.96, energy-to-power-ratio 4, (SDI E3-R135) [69], 5 $t_{CO2}$/MW/a |

**Table A3.** Parameter table on general assumptions.

| Name | C | Capacity/Volume | Count | Parameters |
|---|---|---|---|---|
| **General Inputs** | | | | |
| CO2 | | | 1 | 51.8 Mio. t CO2eq (5% of Germany's energy related emissions of 1990) |
| Natural gas | | Inf | 1 | 26.30 EUR/MWh$_{th}$ [70], 0.2 kg$_{CO2}$/MWh$_{th}$ |
| Hydrogen (H2) | | Inf | 1 | 51.29 EUR/MWh$_{th}$, 0 kg$_{CO2}$/MWh$_{th}$ |
| Biomass | | 193 TWh | 1 | 48.40 EUR/MWh$_{th}$, 0 kg$_{CO2}$/MWh$_{th}$ |
| Transmission grid | | 802 lines, 72 transformers | 575 nodes | New line: 88k EUR/km/a, 4 $t_{CO2}$/km/a; Transformer: 1228 EUR/a/MW, 4 $t_{CO2}$/MW/a (adapted from [33,71,72]) |

## Appendix B. Distribution of CO2 Emissions per Technology

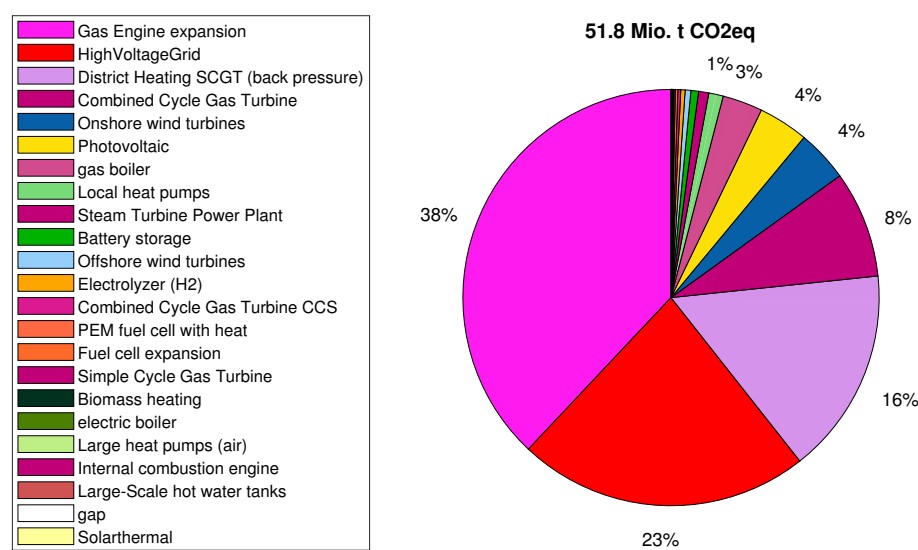

(**a**) 2045+

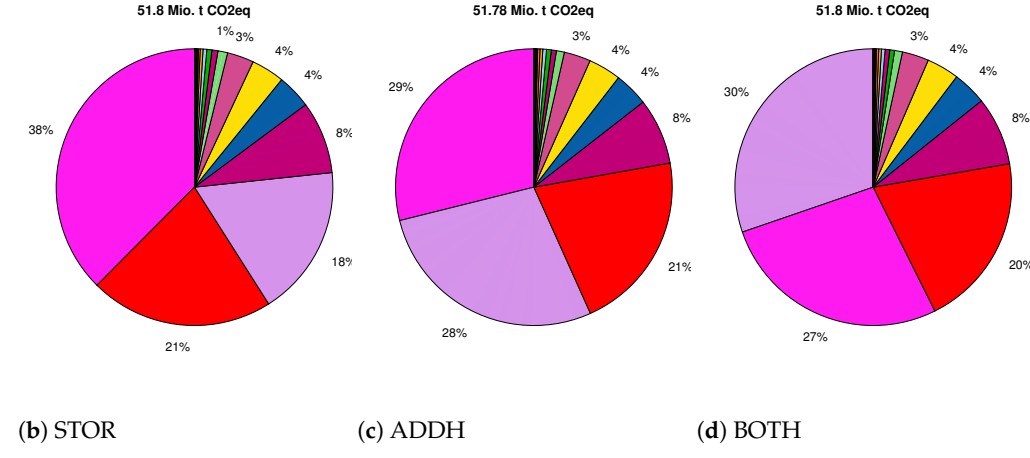

(**b**) STOR                    (**c**) ADDH                    (**d**) BOTH

**Figure A1.** Distribution of emissions (direct and indirect) in t CO2eq per scenario.

## Appendix C. District Heating Technology Expansion

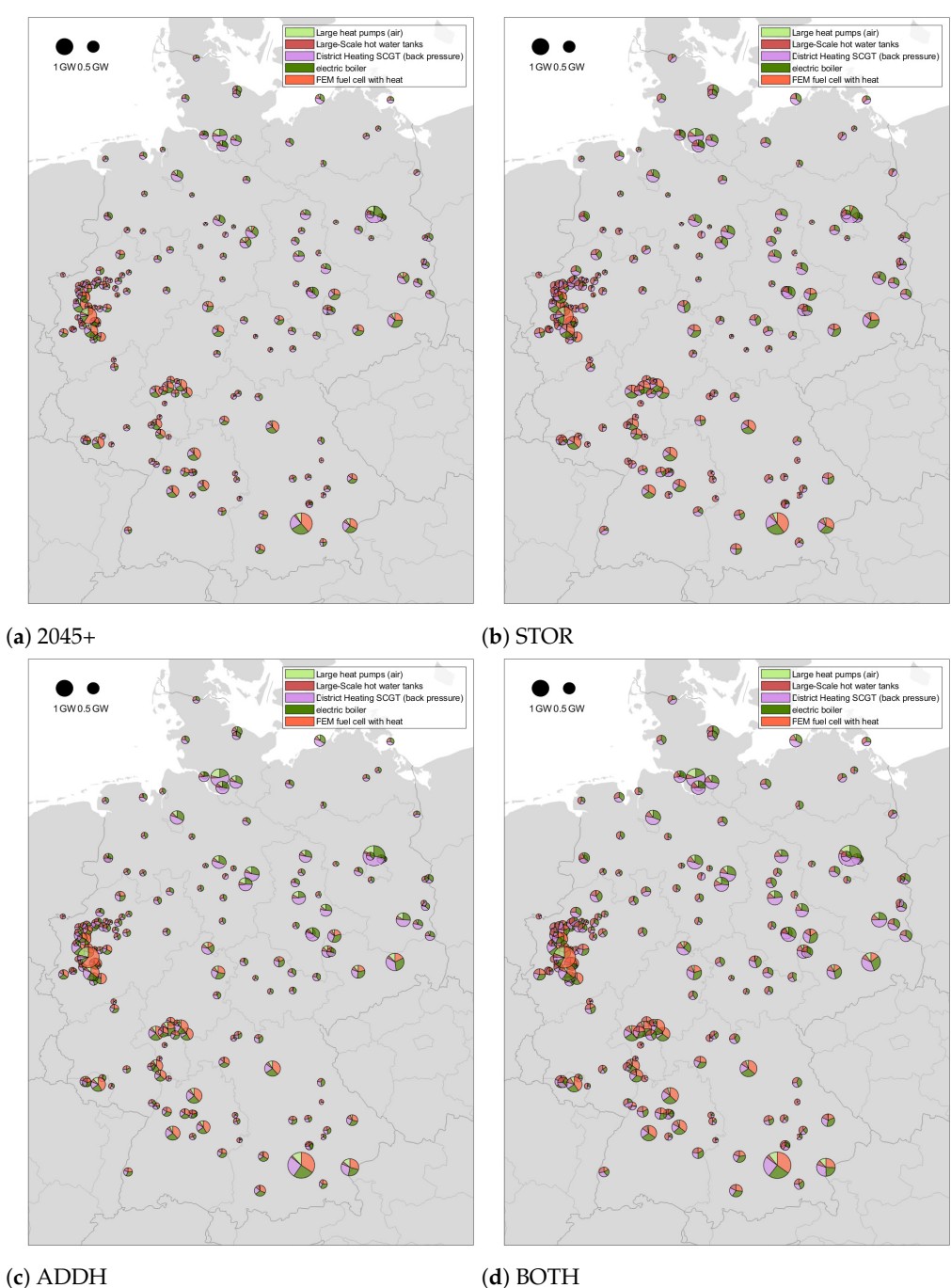

**Figure A2.** Expansion of district heating technologies in the four sensitivities.

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
