# Peer review of "Analyzing Intersectoral Benefits of District Heating in an Integrated Generation and Transmission Expansion Planning Model"

_energies, doi:10.3390/en15072314_

Round 1
Reviewer 1 Report
This paper is to propose analyzes intersectoral benefits of district heating using an integrated generation and transmission expansion planning model. There are several major comments regarding this paper as follows.
(1) Some data, coefficients used to calculate are not described clearly, just giving citations. For example:
L232:“To further not overestimate the potential, the highest spatial resolution was chosen, as suggested in [34].” What is the definition of spatial resolution and the detail value is used in this paper?
L233: " Respective areas were chosen to be modeled as DH expansion areas, marked in red in Figure 2." The description of Figure 2 is not clear. Where is the location? How many districts or cities are chosen to be suitable for DH? Such information will help the readers to better understand the research. Besides, there is no legend of Figure L281:" Based on this, and where applicable, suitable parameters are taken from the study in [36]." There should be a list of parameters and their corresponding values. Besides, Ref.[36] seems not in English version.
L353: "The annual energy demand of commercial buildings is derived using statistical data provided in [54]." Similarly, Ref.[54] is not in English. What kind of data are used? A general description of the data source will help the readers better understand the paper.
(2) The analysis of figures are inadequate. For example:
L304: "The power plant distribution is displayed in Figure 5a." There should be more detailed description, rather than list the figure.
(3) Many figures are Germany's map. However, I suggest giving the border line to distinguish with other country. Also, marking some major cities on the map will help the readers understand better.
(4) The format of references need modification, as some seems not English.
Author Response
Dear reviewer,
we thank you for your intelligible feedback. We have adressed all your comments and did so as follows:
- Coefficients and data
- L232:“To further not overestimate the potential, the highest spatial resolution was chosen, as suggested in [34].” What is the definition of spatial resolution and the detail value is used in this paper?
- We have edited the description of the spatial data set, improved the description of the method and also tried to differentiate more clearly between density definitions. The changes directly referring to your remark:
- L239: "The used data set aggregates the energy demand of an estimated 14million residents into sub-postal code areas, named PLZ8 [35], aggregating to approx. 500 households per code area (detailed data description in Section 2.2.6). It should bementioned that Dochev et al. [36] suggest that for determining district heating applicability, linear heat density (heat demand per pipe length) is actually more accurate than the used areal heat density, but not even approximate data on linear heat density was available at this scale, e.g. heat demand per street length."
- L233: " Respective areas were chosen to be modeled as DH expansion areas, marked in red in Figure 2." The description of Figure 2 is not clear. Where is the location? How many districts or cities are chosen to be suitable for DH? Such information will help the readers to better understand the research. Besides, there is no legend of Figure
- We have improved description of the results as well as the figure and its caption.
- In text we explain L245: "According to Möller and Werner [37], an area is suitable for DH expansion if the annual areal heat demand exceeds 8.3 GWh / km². These areas, marked in red in Figure 2, comprise all urban areas and densely populated suburbs, with little to no potential in the East."
- We have improved the caption of Figure 2 and added city names for better reference, along with a legend.
- L281:" Based on this, and where applicable, suitable parameters are taken from the study in [36]." There should be a list of parameters and their corresponding values. Besides, Ref.[36] seems not in English version.
- We have added a comprehensive of parameters used in Appendix A (page 23 & 24). It comprises two pages of almost all parameters used per submodel, along with colors of the respective technologiy for better overview. Parameters have also been referenced to their sources.
- All non-English sources have been translated in [brackets], in accordance with the MDPI style / ACS citation guidelines.
- L353: "The annual energy demand of commercial buildings is derived using statistical data provided in [54]." Similarly, Ref.[54] is not in English. What kind of data are used? A general description of the data source will help the readers better understand the paper.
- We have improved the description of the data source (as follows) and translated the sourcces name (ACS guideline).
- L379: "The annual energy demand of commercial buildings is derived by intersecting socio-economic properties with statistical data obtained through the extrapolation of survey data, also provided by [56]. Within, the energy demand for lighting, mechanical work, hot water, space and process heating, process cooling and communication infrastructure is recorded for 14 different commercial, retail and service sectors. Using this data, the calculated energy demand depends on the economic department and the estimated number of employees per building."
- L232:“To further not overestimate the potential, the highest spatial resolution was chosen, as suggested in [34].” What is the definition of spatial resolution and the detail value is used in this paper?
- Analysis of figures
- L304: "The power plant distribution is displayed in Figure 5a." There should be more detailed description, rather than list the figure.
- We have enhanced the text of the particular line, but furthermore tried to follow-up whether all key results were conveyed "without being able to see the figure" (I hope this explanation is relatable).
- L324: "The gas power plant distribution and its expansion potential is displayed in Figure 5a, resulting in a huge potential for the expansion of gas power plants in the densely populated areas in western and southern Germany, with only few existing power plants."
- L304: "The power plant distribution is displayed in Figure 5a." There should be more detailed description, rather than list the figure.
- Many figures are Germany's map. However, I suggest giving the border line to distinguish with other country. Also, marking some major cities on the map will help the readers understand better.
-
- I fully agree that the borders are not visible and adding thicker lines would have contributed to better referencing of the locations. With respect to the given time and the number of figures, we were not able to address this: the maps were directly plotted in MATLAB and we found no suitable "basemap" in time to follow your suggestion. However, as noted before, we have added city names to the rather sparse "Figure 2", where it clearly helps in locating the area. We will keep your comment in mind in future works.
-
- The format of references need modification, as some seems not English.
-
- We have modified all references with English translations, according to the ACS citation style (in accordance with the MDPI style guideline).
-
I hope we have been able to respond adequately to all comments and that our improvements are recognizable.
Best regards,
Henrik
Reviewer 2 Report
This paper presents an complex case study on the integration of district heating and power generation systems for an integrated generation and transmission expansion planning model.
This is an interesting subject, namely the combined use of electric and thermal systems. At present time, this could be considered as a source for supplemental flexibility provided by hybrid electric-heat systems.
The paper is well written and organized.
Few remarks should be considered by the authors:
- Page 5, lines 159-161: a simple aggregation of DH demand to one node, neglecting losses associated to the DH networks, could be the cause of significant deviation of computed results as compared to reality. In fact, the authors refer to the issue of losses a little further on page 8, lines 246-247. The authors are invited to present at least a brief analysis of the influence of how to consider losses in the DH network.
- Page 7, Figure 1: text reference to Figure 1 is missing. Add text, with comments to explain the meaning of the representations in Figure 1.
- Page 7, line 222: please check the formulation “… so that basic solutions represent are well parameterized” and, if necessary, rephrase it!
Author Response
Dear reviewer,
thanks for your kind feedback and the intelligible comments. Among other improvements, we have addressed all your comments. In particular:
- Page 5, lines 159-161: a simple aggregation of DH demand to one node, neglecting losses associated to the DH networks, could be the cause of significant deviation of computed results as compared to reality. In fact, the authors refer to the issue of losses a little further on page 8, lines 246-247. The authors are invited to present at least a brief analysis of the influence of how to consider losses in the DH network.
- We have indeed missed the opportunity to talk about losses in further detail and added comments on it in the Method section as well as in the discussion. The relevant additions are:
- L262 (Method) "Heating losses from the network must be added to demand a priori. Heat losses can be described as a function of supply and return temperature, linear heat density and ambient temperature [38 (Chicherin.2020)]. Out of three inputs, only ambient temperature was available. In Approach 2, neither linear nor areal heat densities were used to derive potentials and assumptions about future developments on return and supply temperature were uncertain. It was therefore decided to neglect losses, which was also reasoned by the fact that they were not explicitly modeled with decentral heating options."
- L604: (Discussion) "Another shortcoming is the neglect of heating losses in DHNs. The added heat demand requires more heat production and would presumably reduce the "storage potential", as storage could be used to respond to the heating losses."
- Page 7, Figure 1: text reference to Figure 1 is missing. Add text, with comments to explain the meaning of the representations in Figure 1.
- We have now referenced both parts of Figure 1 [(a) and (b)] and improved the explanation throughout the text. We also changed the captions. Sorry, for not being able to summarize the changes here, as they comprise several lines of minor changes.
- Page 7, line 222: please check the formulation “… so that basic solutions represent are well parameterized” and, if necessary, rephrase it!
- Thanks for pointing this out!
- L225: "A well-known property of linear optimization problems is to determine corner solutions. E.g. in the case of an extraction condensing turbine there are three corner solutions: no output, only electrical output and full heat and electrical output (in the apex of the triangle of Figure 1b). Modeling parameters should be selected in such a way that the corner solutions represent outputs close to reality and with respective efficiency parameters [32]."
I hope we have been able to respond adequately to all comments and that our improvements are recognizable.
Best regards,
Henrik
Round 2
Reviewer 1 Report
The authors have corrected most of the reviewer's concern. I think this can be accepted.